https://doi.org/10.1038/s42003-022-04109-x · **OPEN**
# The complexity of the stream of consciousness

Peter Coppola [1,2], Judith Allanson[2,3], Lorina Naci [4], Ram Adapa [1], Paola Finoia[1,5], Guy B. Williams [2,6], John D. Pickard [2,5,6], Adrian M. Owen[7], David K. Menon [1,6] & Emmanuel A. Stamatakis [1,2✉]

Typical consciousness can be defined as an individual-specific stream of experiences. Modern consciousness research on dynamic functional connectivity uses clustering techniques to create common bases on which to compare different individuals. We propose an alternative approach by combining modern theories of consciousness and insights arising from phenomenology and dynamical systems theory. This approach enables a representation of an individual's connectivity dynamics in an intrinsically-defined, individual-specific landscape. Given the wealth of evidence relating functional connectivity to experiential states, we assume this landscape is a proxy measure of an individual's stream of consciousness. By investigating the properties of this landscape in individuals in different states of consciousness, we show that consciousness is associated with short term transitions that are less predictable, quicker, but, on average, more constant. We also show that temporally-specific connectivity states are less easily describable by network patterns that are distant in time, suggesting a richer space of possible states. We show that the cortex, cerebellum and subcortex all display consciousness-relevant dynamics and discuss the implication of our results in forming a point of contact between dynamical systems interpretations and phenomenology.

[1] Division of Anaesthesia, School of Clinical Medicine, University of Cambridge, Addenbrooke's Hospital, Cambridge, UK. [2] Department of Clinical Neurosciences, School of Clinical Medicine, University of Cambridge, Addenbrooke's Hospital, Cambridge, UK. [3] Department of Neurosciences, Cambridge University Hospitals NHS Foundation, Addenbrooke's Hospital, Cambridge, UK. [4] Trinity College Institute of Neuroscience, School of Psychology, Lloyd Building, Trinity College Dublin, Dublin, Ireland. [5] Division of Neurosurgery, School of Clinical Medicine, University of Cambridge, Addenbrooke's Hospital, Cambridge, UK. [6] Wolfson Brain Imaging Centre, University of Cambridge, Cambridge, UK. [7] The Brain and Mind Institute, Western Interdisciplinary Research Building, N6A 5B7 University of Western Ontario, London, ON, Canada. ✉email: eas46@cam.ac.uk

The modern scientific study of consciousness is marred by a paradox: the objective understanding of a subjective experience. Beyond this, the contemporary neuroscientific endeavour encounters fundamental questions: how do the material and the experiential correspond? How can we compare two subjective experiences? Here we propose an approach to address these two questions.

Contemporary neuroscientific theories have tackled consciousness by proposing that functional relationships among neurophysiological events are fundamental to the emergence of consciousness[1–4]. This may underpin the seemingly unitary nature of experience via the integration of different information (e.g., visual and auditory) and may mediate the dynamic meta-stable re-organisation which is essential to normal brain function and characterises a typical stream of consciousness[5–11].

In functional magnetic resonance imaging (fMRI), "functional connectivity" is often measured as the correlation between blood-oxygenation-level-dependent (BOLD) signals from different brain regions[12]. The functional networks emerging from these correlations have been consistently related to specific mental contents and cognitive states[13–22]. This wealth of evidence suggests that the configuration of brain region connectivity is relevant to specific behavioural contexts and corresponds, at least in part, to subjective phenomenology (i.e., experience) or "mental states". Furthermore, in consonance with theoretical accounts, empirical evidence in the last decade has shown that typical "resting state" functional connectivity networks "disintegrate" in unconscious or sedated states[23–29].

Given the correlation between functional network disintegration and unconsciousness, and between network configuration and cognitive state; in this resting state study we explicitly adopt a framework that assumes the temporal variation of functional network configuration is related to the variations of the vernacular "stream of consciousness". In other words, we assume changes in connectivity patterns correspond, at least in part, to phenomenological fluctuations in mental state and experience (see[30–34] for ontological underpinnings).

In fact, the body of research outlined above has been advanced by characterising different conscious states in terms of the dynamics of their functional networks[25,27,35–38]. These show that the connectivity states visited by healthy conscious subjects are topologically more complex and display rich and structured temporal patterns compared to altered consciousness.

To enable comparisons between conditions and individuals, these studies estimate typically re-occurring connectivity patterns via centroid-based clustering. These studies employ an estimation of different average connectivity states across individual, condition and time, which, although fruitful, can at best approximate a study of a specific individual's dynamic experiential subjectivity. Expressly, by estimating states across all timepoints, clustering techniques discount the temporal unfolding of the stream of consciousness. Furthermore, the comparison of the same connectivity patterns across individuals from highly heterogenous conditions reduces individual subjectivity to an "average dynamical state" of limited interpretability.

To address this gap, we provide an empirical approach that highlights the dynamic neural correlates of an individual's subjective experience that is interpretable in a dynamical systems perspective. We base our approach on the phenomenological observation that any individual experience can be characterised by the experiences anteceding it and the experiences it may lead to[2,9,10,31,39]. Just as any neural state must be understood within its own systemic and dynamic context, we postulate that the intra-individual context of any experiential state is foundational to its subjective value[9,10,40,41]. As mentioned above, we also assume a stable isomorphic mapping between a connectivity and a mental state, by which the difference between two mental states should be proportional to the difference between the corresponding neural states[34].

In dynamic terms[8] a *complex* system can be conceptualised as occupying a specific space in a landscape of many possible states. In a *conscious* system, each state is experientially meaningful due to the subject's modelling of its own past and future in relation to the present[41–44]. Therefore, for each individual, we describe each temporally-specific connectivity state by its similarity to all other available past and future states. We thus obtain an individual-specific, intrinsically defined, temporal landscape via the relationships between all dynamic connectivity patterns. In this manner, we are able to investigate each individual's brain dynamics; specifically, the properties of short-term transitions (~2s) and how informative each state is compared to all others over longer periods of time (>26s). This approach is related to several theoretical concepts that are relevant to the neuroscientific study of consciousness. In accordance with the spatiotemporal theory of consciousness, we focus on the intrinsic time and space constructed by the brain, enabling us to study the consciousness-relevant features of the resulting spatiotemporal landscape[4,45,46]. The present approach is also related to theoretical concepts of the cause-effect repertoire of information integration theory (defined as the repertoire of all possible past and future states given the present state[2]) and entropy (by which consciousness is characterised by high degrees of information specificity and unpredictability; entropic brain hypothesis[1]), to define a proxy measure of the stream of consciousness. Beyond such theoretical antecedents[34] such a landscape has been previously created as an outcome target to evaluate brain computational models under the name of functional connectivity dynamics[6,47,48]. In contrast, the present research will directly analyse the properties of this intrinsically-defined dynamic landscape[4,31] with the object of creating an explicit link between phenomenological and dynamical systems perspectives.

We also apply an analogous approach to diffusion tensor imaging (DTI) data, by which every functional connectivity state is defined intrinsically by its similarity to the underlying structural connectivity pattern. This may indicate how connectivity states vary in relation to the structural connections that underlie it and whether the dynamic relationship between structure and function is affected in unconsciousness.

We hypothesise that individuals in a normal conscious resting state will display specific dynamic characteristics in network transitions (e.g., have higher unpredictability) and will show a more complex landscape or repertoire of states[1,2,4,45]. We analyse four conditions, presented here in order of presumed level of awareness: Healthy Controls: Awake and Moderate propofol sedation; (18 participants[49]; Collected in Cambridge, UK); 11 patients in a minimally conscious state (MCS, collected in Cambridge, UK); and 12 participants with Unresponsive Wakefulness Syndrome (UWS, collected in Cambridge, UK). To ensure results are reproducible, we also use an additional propofol anaesthesia dataset[28], comprised of 16 participants in a control awake, mild sedation and deep sedation conditions. We predict that the measures of intrinsic dynamics will reliably scale with decreasing levels of consciousness[1,2,4,50]. We make use of several brain region definitions (Supplementary Note 1), various pre-processing techniques (Supplementary Note 2) and data to assess convergence of results. If these hypotheses are confirmed at a whole brain level, we will investigate whether similar effects can be differentially explained at a subsystem level (cortex, subcortex and cerebellum). Other than being relevant to theoretical predictions and debates in the literature (e.g.[2,30,51–53]), this latter hypothesis will assess whether anatomically differentiated subsystems (cortex, subcortex and cerebellum) display a temporal

complexity of spatial connectivity patterns which scales with levels of consciousness.

## Results

To obtain a proxy measure of the stream of consciousness, we divided the data spatially into brain regions that covered the whole brain (Supplementary Note 1), and then divided the confound-corrected timeseries (see methods) into different overlapping time windows (24 timepoints moved by 1 timepoint, Fig. 1a; see Supplementary Note 2 for alternative methods used to assess convergence). By correlating the timeseries within each window across brain regions we obtained connectivity matrices that varied in time. Subsequently, we calculated the similarity between each of the temporally-specific connectivity matrices (Fig. 1b) for each individual. We thus obtained a matrix which we called the Meta-Matrix (MM, Fig. 1c), a similarity matrix of connectivity matrices organised linearly in time. This represents a space of intrinsically defined dynamics (Fig. 1d).

**Predictability of intrinsic dynamics**. We sought to investigate the predictability of the intrinsic dynamics of individuals in different states of consciousness. To assess this, we constructed a simple synthetic model (Fig. 1e) in which the similarity between different connectivity patterns decayed monotonically as a function of time. Thus, in this model, timepoints that were further away always displayed less similarity than closer timepoints, effectively modelling a simple dynamic trajectory in state space (e.g., Fig. 1d) that never returns on itself.

We assessed whether the similarity to this simple dynamic model scaled with levels of consciousness (I.e., control awake > moderate sedation> minimally conscious state > unresponsive wakefulness syndrome) by using ordinal logistic regressions (OLR), in which the similarity to the temporal decay of similarity model (TDSM, Fig. 1e) was the predictor and the conditions ordered by presumed level of awareness, the dependent variable.

As expected, unconscious conditions were more similar with the temporal similarity decay model (TSDM) than the more conscious states (Odds Ratio (OR):3.17 $p = 0.0001$ C.I. (2.5%:97.5%) = 1.78:6.22; Fig. 1f). Remarkably, this effect was robust across different TDSM models (linear or exponential decays), parcellations, data and pre-processing pipelines (except when high pass filter was used; see methods and Supplementary Note 3). This may suggest that when a subject is unconscious, their immediate past and future states are more predictably similar to the present state, perhaps indicating sluggishness in short-term connectivity state reconfiguration. Alternatively, this may indicate that there is little similarity of states over longer periods of time in unconsciousness, and thus an altered repertoire of states. Of note is that the TDSM model explained the (MM) intrinsic dynamics of certain UWS patients particularly well (Fig. 1f). Thus, gradual, non-recursive transitions in connectivity patterns poorly described conscious individuals which indicates these may be characterised by short term unpredictable FC reconfiguration, complexity of distal state exploration and potentially intrinsic self-organisation. To further tease apart these possible effects, below we explore the properties of short (2s) and long term (>26s) intrinsic dynamics.

**Proximal temporal complexity**. We investigated to what extent shifts in sequential connectivity states that are temporally close to each other may be predictive of levels of awareness (Fig. 2a). This autocorrelational information is represented in the MM by the values close to the main diagonal (Fig. 2b, in green) which are characterised by high similarity. The distance (i.e., inverse

of similarity; reproduced with other metrics) between each temporally successive connectivity pattern is represented (timepoint 1 to 2; 2 to 3 etc.; see Fig. 1a) in the first subdiagonal (the second longest diagonal, just above/below the main diagonal). On this timeseries of short term connectivity pattern similarities, we calculated measures of central tendency (e.g., median), distribution breadth (e.g., standard deviation), and temporal complexity measures (i.e., sample entropy[54]; see Supplementary Note 4 for alternative methods). We found that, on average, unconsciousness tended to be characterised by higher similarities in short temporal sequences (OR = 4.06, C.I.1.7:9.97, $p = 0.002$). This effect was particularly prominent in deep anaesthesia from the second dataset and in the UWS patients (Fig. 2c; see Supplementary Note 4 for deep anaesthesia data). Thus, intrinsically defined network dynamics had a lower rate of change or speed (less distance/difference over time [2s]) in unconsciousness.

We then investigated the distribution of short-term temporal distances (Fig. 2d). Intriguingly, we found that the standard deviation of proximal time similarity values decreased with levels of awareness (OR = 3.50, C.I. = 2.02:6.60 $p = 0.00001$), in contrast to what would be expected[1]. We interpreted this to signify that consciousness is characterised by smoother, more constant, intrinsic dynamic rates of change (~speed). To confirm this, we took the average of the absolute derivative of the proximal similarity values, indicating how the speed itself varied over time (i.e., whether the rate of change accelerated/decelerated) and found it was highly correlated to the standard deviation of proximal time similarity values (Rho = .84; Supplementary Note 4 for reproductions). Thus, although on average faster, changes in network configurations tend to be, overall, more constant in consciousness (i.e., tend to not accelerate much).

Finally, we found that the temporal complexity (measured via sample entropy[54], reproduced with effort to compress[55]) of proximal timepoint network transitions increased with levels of awareness (OR:4.41, $p = 0.000006$ C.I. 2.37:9.1; Fig. 2e). This indicates that the moment-to-moment transitions of connectivity states are more unpredictable in consciousness.

These results were robust across various measures and control analyses (Supplementary Note 4) and lead to interesting characterisations of the temporally proximal transitions of networks during awareness. Whilst network transitions tend to be faster and have unpredictable temporal sequences, transitions overall tend to be smoother, with more constant rates of change. This suggests that consciousness is associated with network dynamics that are temporally complex, but nonetheless display a certain structure (i.e., more constant transitions).

**Distal dynamic complexity**. Having explored the consciousness-related properties of short-term transitions between different network states, we sought to characterise the wider, distally-defined, state space. Whilst the proximal analyses investigated how networks changed from one timepoint to another, this distal part of the analyses investigates where one connectivity pattern (i.e., a column in the MM) sits in relation to all other individual specific states (Fig. 3a).

To ensure proximal timepoints did not influence results, we took the *off-diagonal triangle* of the MM that represents similarities between connectivity patterns that are distal in time (Fig. 3b). By removing proximal timepoint autocorrelations, we are able to investigate the complexity of an individual's connectivity position in state space over longer periods of time. We thus removed from the MM the 13 timepoints (26s) closest to each state as this meant that more than half of the sliding window was not overlapping (Fig. 1a). We also repeated the analyses with

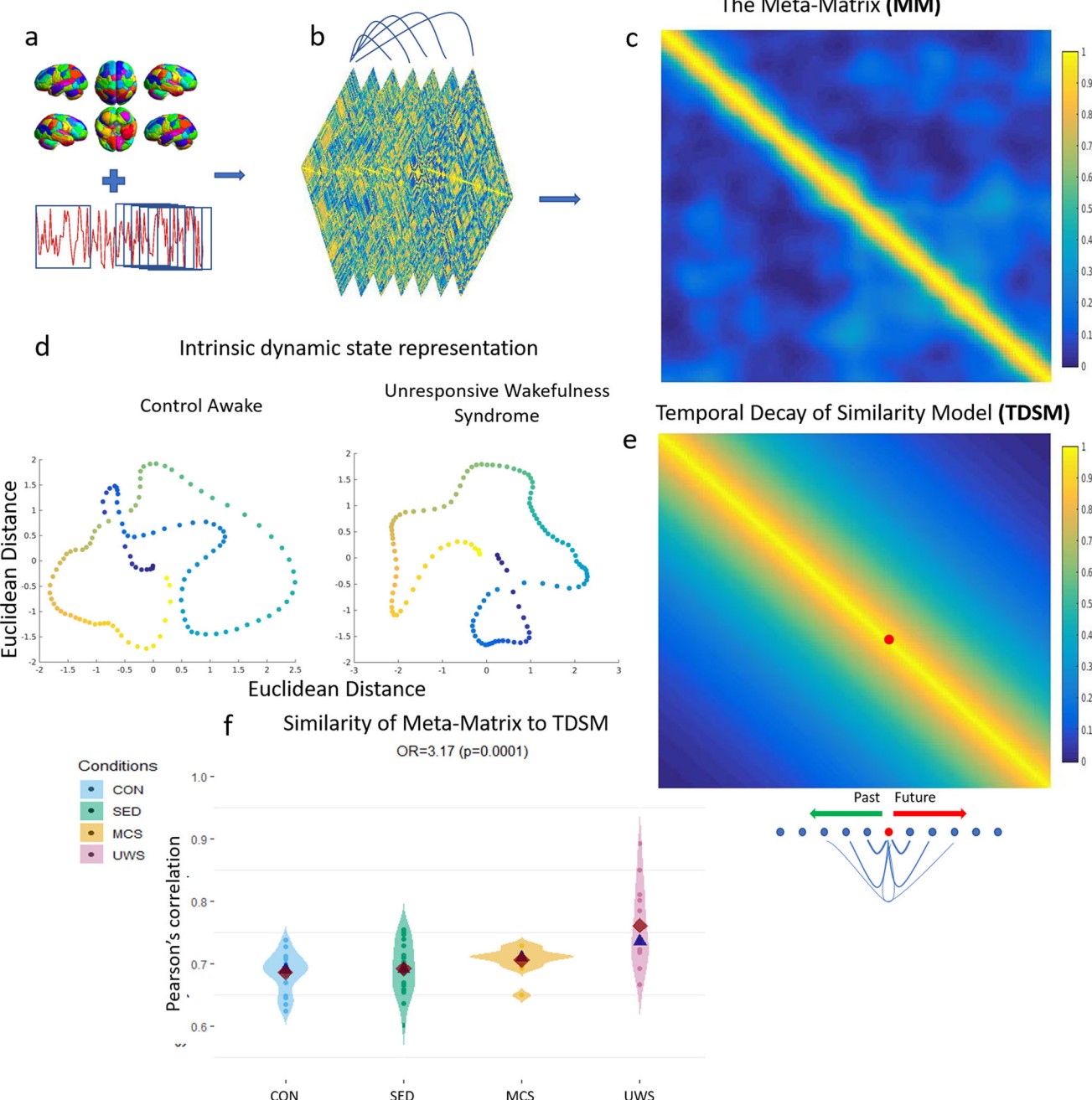

**Fig. 1 Meta-Matrix formation and predictability of intrinsic dynamics. a** Shows the methodological approach we adopted in this study. We parcellate the data spatially and then temporally using the sliding window approach, where one window is comprised of 24 timepoints and is moved forward by one timepoint. By correlating all parcellated brain regions within each widow we obtain time-varying connectivity matrices (**b**; see Supplementary Note 2 for alternative methods). By comparing all connectivity matrices to all other sampled ones (by vectorising the upper triangle and correlating them via Pearson's correlation) we obtain the Meta-Matrix **c**, also known as functional connectivity dynamics[47]. Each column in the meta matrix represents one connectivity matrix and all the cells within that column represent its similarity to all available past and future connectivity states. Thus, each connectivity pattern is described by its intrinsic relationship to all other connectivity states. By calculating Euclidean distance on these relationships and reducing their dimensions, we can represent the MM on a 2D plane **d**. In **d**, each point represents a connectivity state and its vicinity to other points represent its similarity. This may be interpreted as a representation of an individual moving through possible connectivity states as the scan progresses (hotter colour = later in the scan). Noticeable is that the UWS has consecutive states that are closer (more similar to each other). We tested the predictability of the whole MM by comparing the meta-matrix to the temporal decay of similarity model (TDSM) **e**, in which timepoints that are closer to each other are more similar (three of these models constructed, see methods; first exponential decay model presented in this figure). **f** We find that the similarity of the meta matrix to the TDSM predicts increasing levels of awareness (control awake > sedation > minimally conscious state > unresponsive wakefulness syndrome) (Odds Ratio = 3.17, p = 0.0001). TDSM = temporal decay of similarity model, Con = control awake n = 18, SED = sedation n = 18; MCS = minimally conscious state n = 11; UWS = unresponsive wakefulness syndrome n = 12. Red rhombi represent the mean, whilst blue triangles, the median.

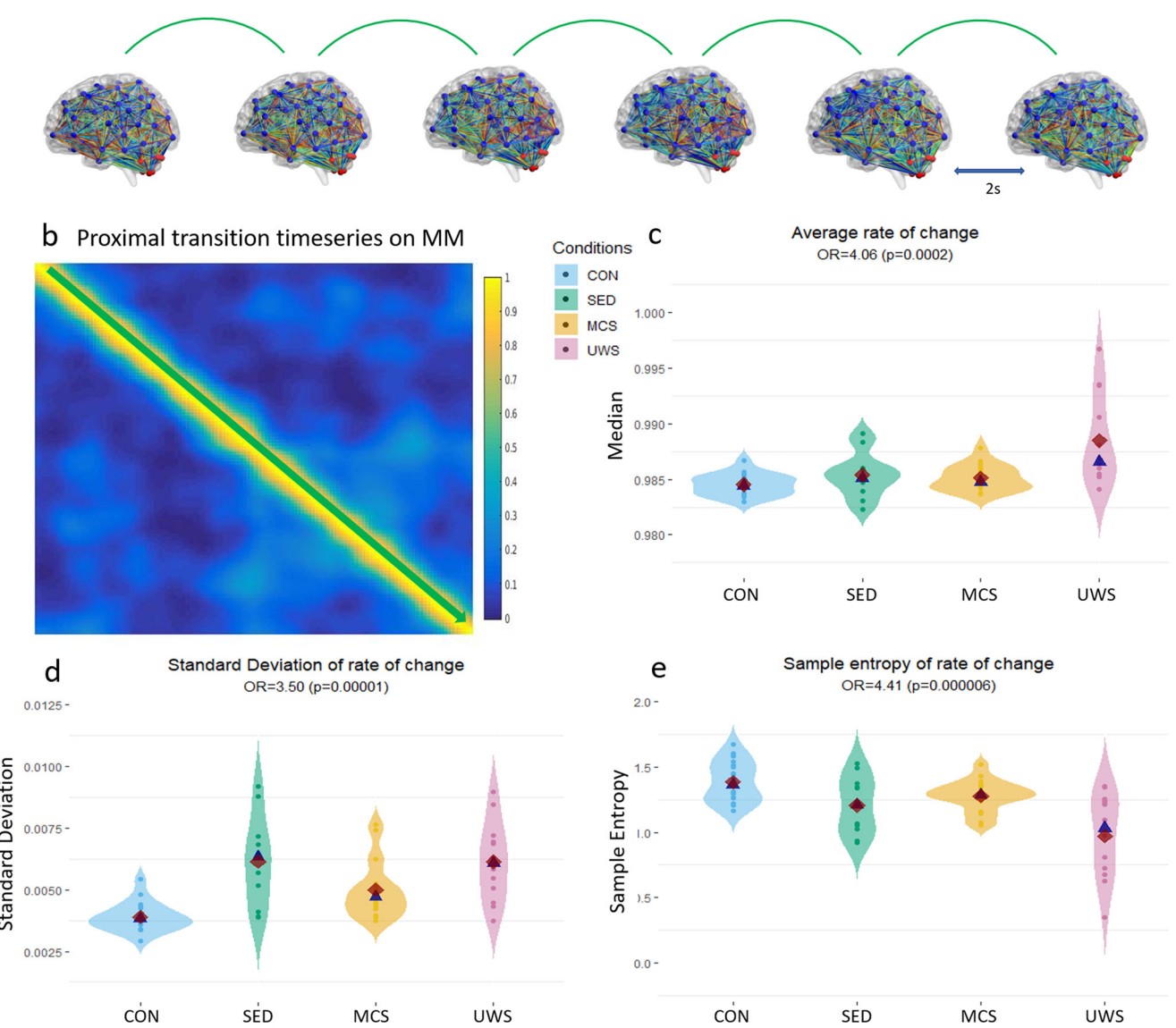

**Fig. 2 Proximal network transitions; description and analyses. a** Depicts the short-term transitions between each successive connectivity pattern (time = 2s). The distance between these was measured via Pearson's correlation (see Supplementary Note 2 for alternative distance metrics used). **b** Shows where in the meta-matrix this information is represented (along the diagonal); shown by a green arrow. On the resulting timeseries, describing the rate of change of connectivity patterns over time, we calculated the average **c**, the standard deviation **d** and the sample entropy (**e**; see Supplementary Note 4 for alternative methods). These measures were inserted as an independent variable in ordinal logistic regression with conditions ordered according to presumed levels of awareness (control awake > sedation > minimally conscious state > unresponsive wakefulness syndrome). CON = control awake n = 18, SED = sedation n = 18; MCS = minimally conscious state n = 11; UWS = unresponsive wakefulness syndrome n = 12. OR = odds ratio. Red rhombi represent the mean, whilst blue triangles, the median.

a 24-point cut-off (48s) as this ensured sliding windows were not overlapping. We then created another symmetrical matrix from the off-diagonal triangle by mirroring it across the new diagonal (Fig. 3b, c) and obtained the distal meta-matrix (dMM; Fig. 3c). Each column of this new matrix represents a connectivity pattern and the cells represent the similarity to other connectivity patterns that are distant in time. The dMM (Fig. 3c) therefore approximates the position of a specific connectivity state in a wider space of possible states, via the relationship (i.e., distance) to all other states. Whilst the proximal space dynamics are more easily understood (as time progresses, similarity tends to decay; or distance tends to increase [see TSDM Fig. 1e]), the dMM may approximate a more complex and higher dimensional state space

(e.g., see Fig. 1d; see Supplementary Note 5 for further characterisations). The difference between the proximal and distal space is corroborated by the difference in similarity values and their organisation (see differences in colour in different parts of the MM (e.g., Fig. 3b), also compare the TDSM (Fig. 1e) to the dMM (Fig. 3b, c)). Interestingly, the average maximum similarity found in the dMM across individuals was r = 0.37 (standard deviation = 0.092; for a higher granularity parcellation, mean maximum r = 0.26, standard deviation = 0.089).

We thus sought to investigate how properties of the dMM scale with levels of consciousness. We found that there was no appreciable difference in the average and variation of the similarity values represented in the dMM (Supplementary

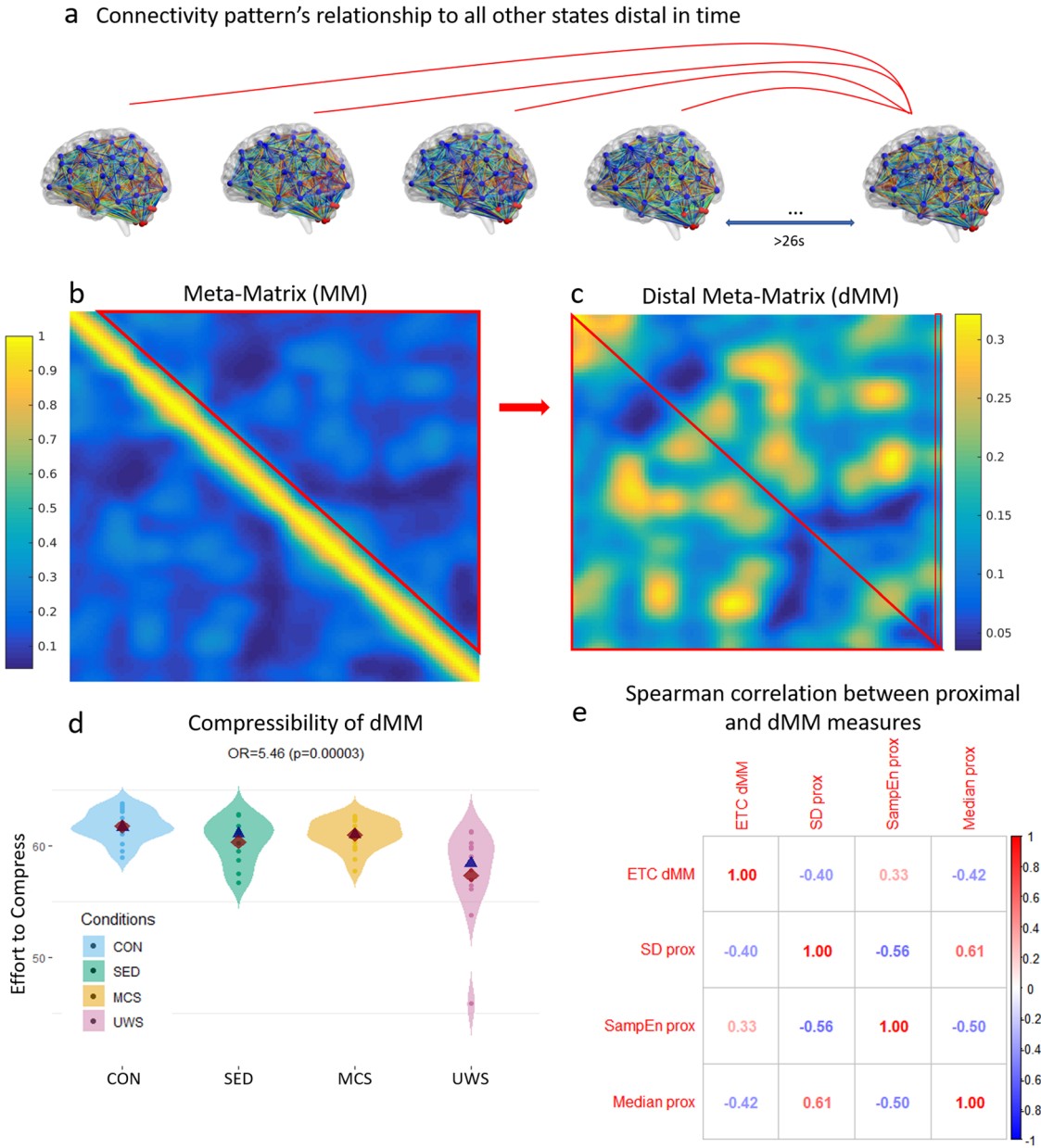

**Fig. 3 Distal meta-matrix description and analyses. a** Shows the relationship (i.e., distance) of one connectivity pattern to all others that are distal in time. By taking the off-diagonal values of the meta-matrix (represented by a red triangle in **b**) and mirroring it across the new diagonal we obtain the distal meta-matrix (dMM; **c**). This represents relationships (Pearson correlation) between connectivity patterns that are distant in time (distances of 13 and 24 timepoints were chosen, as these corresponded to more than half and the entire sliding window [Fig. 1a] respectively; latter distance is displayed here), therefore providing a metric of the definition of connectivity states over longer periods of time. We calculated complexity measures (ETC and sample entropy) on each column of the distal meta-matrix and averaged across these (represented by the red rectangle in **c**; see Supplementary Note 6 for reproduction). This average complexity measure was inserted in an ordinal logistic regression (**d**) with conditions ordered according to presumed levels of awareness as the dependent variable (control awake > sedation > minimally conscious state > unresponsive wakefulness syndrome; 13 proximal timepoints removed in this instance). **e** Shows the shared variance between the distal and proximal measures (spearman correlation; p < 0.001). dMM = distal meta-matrix, Con = control awake n = 18, SED = sedation n = 18, MCS = minimally conscious state n = 11, UWS = unresponsive wakefulness syndrome n = 12; OR = odds ratio, ETC = Effort-to-compress, SampEn = Sample entropy, SD = standard deviation, prox = of proximal temporal transitions. Red rhombi represent the mean, whilst blue triangles, the median.

Note 6). This suggests that with consciousness there is no effect on the tendency to return to similar states, or in the variations of distal definitions of a state. Conversely, when we examined the compressibility of the vectors that distally define each state (columns of the dMM, then averaged; Fig. 3c), we found a robust scaling with levels of awareness (OR = 5.5, CI = 2.56:12.98, p = 0.00004; Supplementary Note 6). This measure approximates how complex a state is on average as defined by

its position (relative distance to its distal past and future) in the wider space of possible states. We take this to mean that states as defined within their intrinsic dynamic space (distally defined; dMM) are less easily describable in awareness. This may be evidence that the position of a state within its dynamic landscape (i.e., "state space" or "phase space" in dynamical systems theory) becomes more complex with the emergence of consciousness.

**Table 1 Results for proximal and distal measures for the cortex, subcortex and cerebellum.**

| Measures | Cortex | | Subcortex | | Cerebellum | |
|---|---|---|---|---|---|---|
| | Odds Ratio | P Value | Odds Ratio | P Value | Odds Ratio | P Value |
| Proximal SampEN | 3.24 | 0.000161 | 2.55 | 0.000015 | 2.58 | 0.09 |
| Proximal STD | 3.42 | 0.000031 | 2.73 | 0.000005 | 3.58 | 0.05 |
| Proximal Median | 3.67 | 0.000975 | 3.53 | 0.000028 | 2.76 | 0.07 |
| dMM ETC | 6.60 | 0.000002 | 3.98 | 0.000013 | 2.14 | 0.09 |

Presented are Odds Ratios, and P-Values for each measure (rows) and each system (columns). *ETC* effort-to-compress, *STD* standard deviation.

**Relationship between distal and proximal measures**. We found that with consciousness, proximal transitions tend to be less predictable, quicker, but, on average, more constant. Furthermore, the wider dynamic state space (approximated by the dMM), seems to be more complex ("less compressible") in consciousness. We investigated to what extent these measures held unique variance. We found that, all variables were correlated to each other ($p < 0.001$, Fig. 3e), although there seemed to be evidence the different measures displayed some independent variance (averaged explained variance 22%), specifically between proximal (Fig. 2c, d, e) and distal measures (Fig. 3d; see Supplementary Note 7 for reproduction). We then inserted the variables with the highest effect sizes (odds ratios) for the proximal and distal measures into the same ordinal logistic regression as covariates (namely sample entropy of proximal transitions and compressibility of dMM). We found that, there was evidence each of these had independent predictive power (dMM compressibility OR = 3.82, $p = 0.0004$; Proximal SD OR = 1.93, $p = 0.02$), although this did not reproduce unequivocally across different control analyses (Supplementary Note 7 for details). We therefore cannot conclude whether proximal and distal measures have unique predictive power.

**MM measures in the cortex, subcortex and cerebellum**. We then investigated whether any of the intrinsic dynamic effects (proximal and distal, Figs. 2 and 3) can be ascribed to the cortex, subcortex and cerebellum individually. We found the cortex and the subcortex tended to have high effect sizes for all measures, whilst the cerebellum was only liminally significant (see Table 1). Remarkably, the subcortical effects reproduced across all control analyses, whilst the cortex did not reproduce with global signal regression in the proximal measures (Supplementary Note 8 for details). We then sought to investigate whether these cytoarchitectonically distinct systems had unique predictive power. We therefore inserted the intrinsic dynamic properties of the cortex, subcortex and cerebellum as covariates in the same ordinal logistic regression. We ran a separate ordinal logistic regression for each measure, namely the average, standard deviation and sample entropy for the proximal transitions and the compressibility of the dMM.

We found that the only effect that consistently reproduced across analyses (described in Supplementary Note 9) was the dMM compressibility of the cortex, whilst the significance of the temporal properties of the subcortex and cerebellum seemed to depend of pre-processing contingencies (Supplementary Note 2 for description of preprocessing contingencies; and Supplementary Note 9 for results).

**Connectivity pattern dynamics in relation to structural connectivity**. We sought to further investigate brain dynamics in consciousness via the intrinsic relationship between dynamic functional connectivity and static structural connectivity (measured via tractography of diffusion tensor imaging). For this analysis, instead of comparing each functional connectivity pattern to all functional patterns available, we compared each of the temporally-specific functional connectivity patterns to the same individual's structural pattern (which were parcellated using the same region definitions). We thus obtain a timeseries of how similar each connectivity pattern was to the structural pattern (Fig. 4a). On this timeseries, we calculated the sample entropy and effort to compress. Similarly, to the above analyses, we inserted the complexity measures into an OLR as a predictor with the ordered dependent variable composed of a control condition, minimally conscious state, and unresponsive wakefulness syndrome respectively. This enables us to investigate how the dynamic relationship between structural and functional connectivity relates to different levels of consciousness.

For the whole brain, the relationship between structural and dynamic functional connectivity was more complex (across sample entropy and effort to compress measures) in higher levels of consciousness (OR = 2.95, $p = 0.002$ C.I. = 1.39:6.27; Fig. 4b). This result reproduced across various analysis permutations and parcellations (Supplementary Note 10) and suggests that functional connectivity patterns have more freedom to vary in relation to the structural connectivity that is presumed to underpin it. In contrast to previous research[35], we do not cluster across functional states, but consider the unadulterated functional dynamics of each individual and characterise it by comparison with the individual-specific structural connectome. If we take the assumption that, to an unknown extent, connectivity states correspond to phenomenological states, these results may be taken to indicate the increased possibility of phenomenological variability that arises with consciousness.

We then tested whether a similar effect is to be found using the structural and dynamic functional connectivity of cortex, subcortex, and cerebellum parcellations. The complexity of the relationship between dynamic functional and structural connectivity did indeed increase with levels of consciousness in the cortex (OR = 5.42, $p = 0.003$; C.I. = 2.05:14.35). The subcortex (OR = 3.84, $p = 0.0004$, C.I. = 1.72:8.51) and the cerebellum (OR = 3.36, $p = 0.001$, C.I. = 1.49:7.53) showed the same pattern (Fig. 4b). Interestingly, when the complexity of the structure-function dynamic relationship for each subsystem was inserted in the same OLR, both the cortex ($p = 0.01$) and the subcortex ($p = 0.0003$) seemed to retain independent predictive power, whilst the cerebellum was not significant ($p = 0.07$; see Supplementary Notes 10, 11 for various reproductions). These analyses show that complex structure-function dynamics are a characteristic of different subsystems in consciousness, and provide important evidence for debates in the literature.

Furthermore, we tested whether there were any differences in the maximum similarity between structural and functional connectivity patterns. We found that the DOC conditions were significantly different from the control condition for the whole brain (Con > MCS Z = −2.17, p = 0.029; Con > UWS: Z = −2.09,

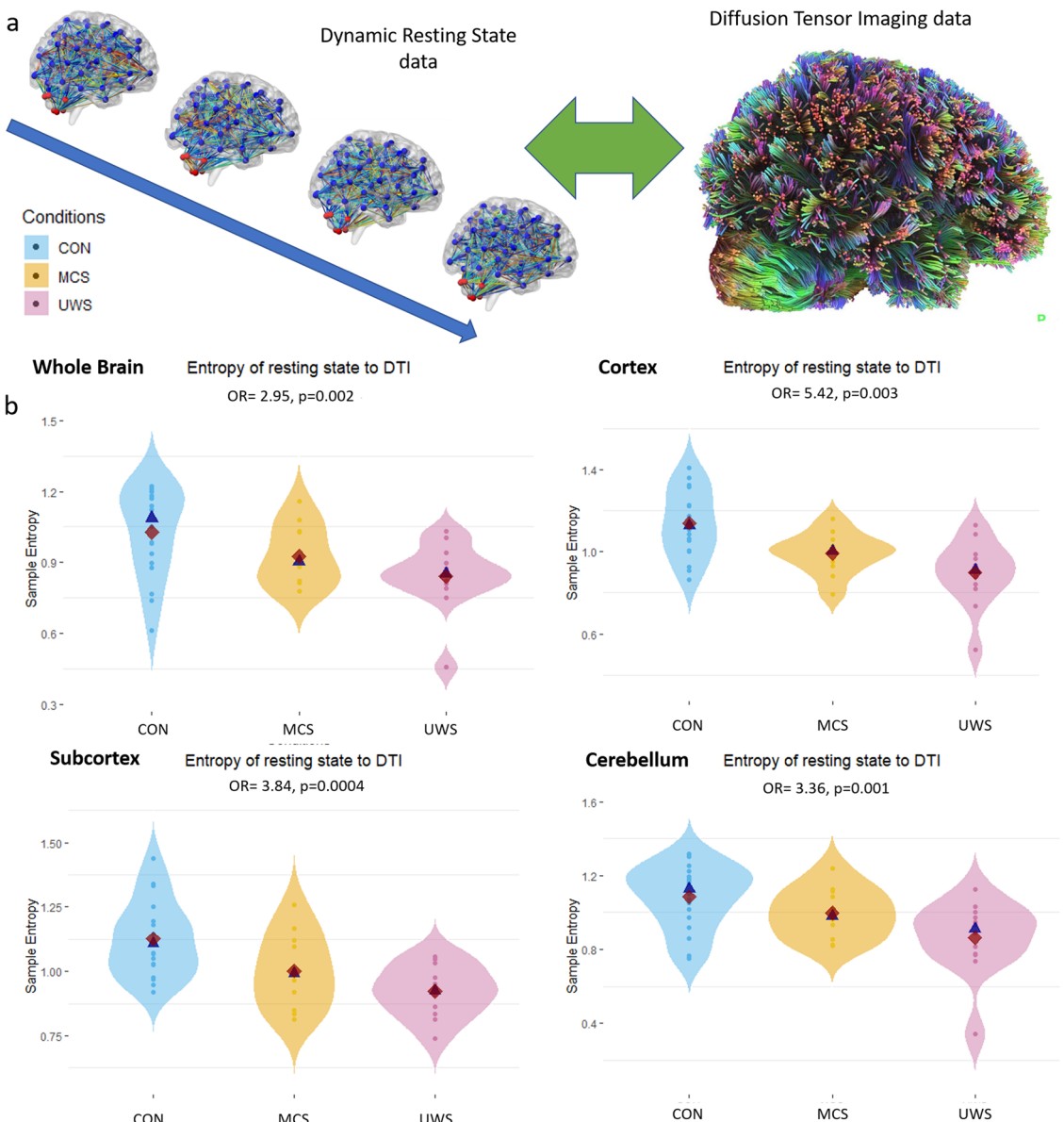

**Fig. 4 Complexity of structure-function dynamic relationship.** Illustration of method (a) for the sample entropy of the relationship between functional dynamic and tractography of diffusion tensor imaging data. We measured the similarity (using Pearson's correlation) of each dynamic functional connectivity state to that of the underlying structural connectivity (measured via tractography). From this we obtained a string of similarity values on which complexity measures (sample entropy and effort to compress) were calculated. This value was then inserted as the predictor variable into an ordinal logistic regression. Conditions, ordered according to presumed level of awareness (control awake > minimally conscious state> unresponsive wakefulness syndrome), were the dependent variable (b). Shown are the results for the whole brain, the cortex, the subcortex and cerebellum. DTI = diffusion tensor imaging, Con = control awake n = 18, MCS = minimally conscious state n = 11, UWS = unresponsive wakefulness syndrome n = 12 (n = 11 for cerebellar results). OLR = ordinal logistic regression. Red diamonds represent the mean while blue triangles the median.

$p = 0.036$). However, cortex (Con > MCS: $Z = -1.73$ $p = 0.083$, CON > UWS: $Z = -2.065$, $p = 0.038$) and the subcortex (Con > MCS: $Z = 0.4719$, $p = 0.6370$, Con > UWS: $Z = -2.5765$, $p = 0.0100$) displayed differences only for UWS. Interestingly, the subcortical maximum similarity between structural and functional connectivity was close to significant when comparing UWS and MCS ($z = -1.93$, $p = 0.0525$). However, these tests do not survive multiple comparison correction.

These results suggest that, while functional connectivity patterns of conscious participants are characterised by greater variations in relation to the structural connectivity, DOC patients may display higher similarity to the structural patterns and less variations over time according to severity. To the best of our knowledge, we show for the first time, that the complexity of functional-structural dynamics of the cerebellum, the subcortex and cortex, scale monotonically with level of awareness in DOC.

Thus, the functional connectivity dynamics of consciousness are characterised by a greater freedom, and are able to depart from the underlying structural connectivity as experiences unfold over time.

## Discussion

Our results show that the network dynamics of awareness display specific characteristics. In consciousness, transitions between connectivity states were on average faster, yet overall maintain a

more constant speed. We found short term transitions were also more unpredictable, whilst long term intrinsic descriptions of a connectivity state were more complex (i.e., 'hard to summarise'). We also found a higher degree of temporal complexity in the intrinsic dynamic relationship between functional connectivity state to their structural underpinnings, implying greater freedom of functional variation in relation to the structural backbone. Given the consistent relationship between network connectivity and cognitive states[13–22] these specific temporal network properties may be related to phenomenological dynamics of mental states and content[1,2,4].

The intrinsic temporal organisation of neural activity has been thought to provide a fundamental scaffold on which unified experience may emerge[4,56–58]. Previous studies have in fact showed a disruption of intrinsic temporal structure (e.g., self-similarity) of BOLD and electroencephalogram signals in different states of unconsciousness[37,56,57,59,60]. Here we show that in unconsciousness, temporal organisation is affected also at the network level in specific ways (i.e., with slower average speed but relatively more acceleration and deceleration, with more predictable transitions and a less complex state space). The temporo-spatial theory of consciousness[4] postulates that the intrinsic temporal-spatial organisation of the brain is fundamental to the emergence of consciousness. In accordance to this theory's predictions, we show that consciousness is associated with specific autocorrelation properties and a more complex dynamic repertoire which would theoretically support subjective feelings of continuity and fluctuations of states of consciousness over time[45]. Similarly, the approach set forth in this paper may permit to approximate an empirical measure of the size "cause effect repertoire" in Information Integration Theory terms, (i.e., the distribution of possible past and future states given the present state; thus defining "information" in this theory[2]). Of particular interest is that whilst increased complexity of (proximal and distal) temporal sequences is in line with the entropic brain hypothesis (stating that consciousness leads to an increase in entropy[1]), the increased breadth of proximal transition distances (which reproduced with Shannon entropy, see Supplementary Note 4) would superficially seem to contrast this theory (see also[45]). However, a deeper reading of this theory reveals that consciousness is acknowledged to be associated with an organisational effect which ultimately functions to reduce "surprising" states of the organism at different levels (biological, psychological, see[30,44,61]). Therefore, the increased moment-to-moment unpredictability, but reduced variations in network transition speeds, may represent functional organisational properties that emerge with consciousness.

Higher temporal complexity of connectivity patterns over longer periods of time are in consonance with a series of other recent papers[5,25,27,37,38,62] which report an increased richness of repertoire of cortical states visited during consciousness in consonance with theory[1,2,4]. However, these studies all use clustering approaches, which, beyond potential methodological issues (e.g.[63]), rely on discrete averaged states that are estimated across individuals and conditions of awareness. This approach, although successful, denatures the individual-specific temporal unfolding of experience and neural dynamics, and often requires inverse inference (e.g.[27]). In fact, we found relatively low absolute similarity in the dMM, perhaps suggesting dynamics are perhaps best represented along continuums rather than by categorical assignment (for e.g., see Shine and colleagues[21]). Nonetheless, we advance this body research by showing that temporally linear, intrinsically-defined, non-clustered spatial connectivity patterns show specific dynamic characteristics in awareness, both in both proximal and distal time-points.

Due to the fact that the MM may represent an individual's exploration of a space of possible states, the present results also

lend themselves to a dynamic systems interpretation. The lower dMM compressibility, the characteristics of dynamic connectivity patterns in short term transitions, and the untethering of the functional from the structural connectome may reflect differences in the underling landscape describing possible state trajectories. More specifically, reduced variation in speed may indicate that the underlying dynamic landscape of consciousness tends to be flatter overall, as shallower "basins" (i.e., attractors) would result in less acceleration. Nonetheless, the unpredictability of short-term transitions may indicate a more complex and detailed local topology. Furthermore, the incompressibility of distally defined states and the increased structure-function dynamical complexity may perhaps indicate a higher dimensional, more complex, state space.

In fact, Deco and Kringelbach[6] suggest that the MM contains information about the metastability of the system (the [in]stability of system state changes over time[8]). The pharmacological and lesion perturbations characterising the data most likely influence the dynamic landscape of possible functional states making transitions slower, less constant and more predictable. Nonetheless, other studies suggest that the functional dynamics of typical wakefulness is characterised by structured spatial-temporal circuits[16,21,27,64,65], which are discriminative between different conscious[5,27] and cognitive states[14,15,20]. This suggests that state transitions in consciousness, are not only complex, but also structured (as found in this study), and predicated by the context and current and previous states. Thus, with the emergence of complex awareness, additional "apparent" changes[7] in the dynamic landscape may arise due to the structured, relative trajectory or "perspective" of the current brain state within its own intrinsic space of past and future states[4,42,43,45,66]. Such an interpretation of neural dynamics can be seen as intuitively complementary to phenomenological and psychological dynamics[9,10,30,61,67,68], by which the specific (psychological) state of a conscious system will determine its possible future states. In other words, the "perspective" (positioning and direction; see[31]) of a brain state within its intrinsic dynamic landscape would in part correspond to a complementary psychological-experiential "perspective". This "dual-aspect" perspective can be potentially identified with the system's hierarchical inferences on its own past and future states which imbue the present with subjective meaning[43] and determine its dynamics[2]. We tentatively propose that the correspondence between network dynamics and phenomenological unfolding may be considered a mapping between mind and brain (for ontological elaborations see[30,33,34,45,52]).

We also expand on previous dynamic functional connectivity studies[25,27,36–38,62] by incorporating analyses that investigate how the relationship between anatomical and functional connectivity changes with different levels of awareness. We show that, in relation to the individual specific anatomical connectivity, non-clustered functional patterns have more freedom to vary in higher levels of awareness. This complex relationship is thought to be a fundamental characteristic in normal conscious brain dynamics in that it allows for an extensive repertoire of possible states and therefore enables adaptation to the internal and external environment[47,48,69]. In fact, previous studies, looking at anaesthesia in macaque monkeys and rats[35,70] and deep sleep in humans[71], show that cluster-defined or temporally-averaged functional connectivity states become more similar to anatomical connectivity during anaesthesia. We build on these results via a novel technique that does not rely on clustering and show that the complexity of the intrinsic relationship between functional and anatomical connectivity decreases in humans according to severity of DOC diagnosis, thus tentatively providing evidence of phenomenological value intended for such diagnoses[72]. We also show, for the first time, that this consciousness-dependent dynamic relationship

between functional and structural connectomes exists in the sub-cortex and cerebellum as well as the cortex. These results can be interpreted two ways. If DOCs are to be considered cortico-thalamic disconnection syndromes (e.g.[73,74]), then white matter pathology may be causing a mechanistic impairment to the breadth of upstream cortical functional configurations[48,69]. However, changes in the relationship between anatomical and functional connectivity have been found in anaesthesia[35,70], psychedelics[75], and deep sleep[71]. Furthermore, these effects are found in the subcortex and cerebellum beyond the cortex. Therefore, it is plausible the these results are partly driven by a decrease in the experientially-driven dynamic self-organisation that would emerge with consciousness[1,2,4,5,69]. Either way, unconsciousness is char-acterised by the lack of possibilities to occupy different states depending on previous or possible future (biological and experi-ential) states (refer to "surprise"[44] and "temporal thickness"[42]). Therefore, expanding on the dynamical systems interpretation above, we speculate that in unconsciousness, dynamic self-organisation will be tendentially driven by fixed biological con-straints, (such as metabolism and white matter structure), rather than being additionally driven by the contextual phenomenological experience of the organism.

Another contribution of this study is that, whilst previous dynamic functional connectivity studies focused on the cortex[5,25,27,35,37,62], we performed dynamic analyses on the whole brain, and three subdivisions of it (cortex, subcortex and cerebellum). The neural correlates of consciousness are often thought to be localised in the cortex[2,5]. In fact, we show that the cortex's dMM compressibility has consistent unique predictive power. Nonetheless, we also show that the subcortex's temporal properties are also consistently predictive of levels of con-sciousness when considered on its own. Given the fundamental role of subcortical structures in homoeostasis and fundamental subjective feelings (e.g., hunger, fear)[52,76] and the synergistic nature of brain function, we speculate that the consciousness-specific dynamical properties found in the present research may be in part explained through subcortical tonic and phasic neuromodulation[21,77–79]. Furthermore, the findings that the cerebellum does show some consciousness-relevant variations may be controversial. In fact, IIT initially stated that the cere-bellum is not important for consciousness due to the lack of a coexistence of functional specialisation, integration and feed-back mechanisms[80]. Other work[2,36] cites clinical cases of cer-ebellar agenesis as evidence that its presence is not a necessary condition of consciousness. Conversely however, specific clin-ical cases of Hydranencephaly show that experiential content may emerge without most of the cortex[53,78]. For example, a paper describes four children defined as "decorticate" (and therefore prognosticated with permanent UWS) that displayed signs of affect, perceptual discrimination, social interaction, play, aesthetic enjoyment etc.[81]. These authors suggest the fortunate social-developmental context of these specific indi-viduals may have made all the difference to the emergence of awareness. Despite claims in consciousness research[2,82,83], there is in fact substantial evidence for the complex organisa-tion of the cerebellum[64,84], that it displays consciousness rele-vant variations (e.g.[85]), and for its role in functions that are relevant to contents of awareness such as language, perception and emotion[86–90]. Although the cerebellum may not typically be a necessary or sufficient condition for consciousness, it may be hypothesised that the spatial temporal re-organisation of a conscious system would contingently influence and be influ-enced by all parts related to information content. Therefore, as evidenced by this paper, it is likely that the (typically devel-oped) cerebellum has a role in shaping the stream of consciousness.

**Methodological considerations.** The meta-matrix approach has been discussed before under the name of "functional connectivity dynamics"[6,47,48], however as opposed to the present phenom-enological approach, the original creators used the MM as a target of computational modelling of whole brain dynamics. This work praises the sensitivity to dynamics this approach affords, and given the reproducibility of our results (see Supplementary Notes 1, 2 and methods for description) we corroborate this. The present approach, which was partly inspired by representational similarity analysis[91] and is analogous to topological data analysis[92,93], enables a description of an individual-specific brain state space in which each state's relationship to all others may be represented. Furthermore, Battaglia and colleagues[94] developed a similar method of speed under a different framework to investi-gate how ageing relates to functional network dynamics. It is possible, with greater amounts of temporal data (e.g., electro-encephalogram) that methods such as these may permit the study of the structure of an individual's stream of consciousness in case studies and across specific experiential states. Such an approach also may permit the study of a specific connectivity pattern (e.g., taking one column of the meta-matrix) by its relationship to all others, or the specific differences between two sequential states (by taking the difference between the connectivity pattern). Although the present study tried to intrinsically represent the stream of consciousness, we could not characterise the content of the stream of consciousness as each connectivity pattern was defined intrinsically via its similarity to all other patterns in resting state data. The use of online experience sampling[18] or naturalistic paradigms (e.g., audio, music, or movie watching[28], may provide a specification of the content of an individual's stream of consciousness. Furthermore, uniting the present approach with clustering techniques[21,25,27] may permit dynami-cal systems characterisations of specific (clustering-defined) states (e.g., acceleration towards a DMN state).

In conclusion, we take a phenomenological approach to show that consciousness is characterised by a temporal unfolding of connectivity states that is quicker, stable (without much acceleration/deceleration) and more unpredictable. We also show that connectivity states are less easily described by their relationship to distal timepoints and have more freedom to vary in relation to their structural backbone. The present approach may provide a principled empirical way to characterise the perching and flights of the "stream of consciousness" (Fig. 1d) as metaphorically described by William James over a century ago[10]. We tentatively propose that this difference is in part driven by additional emergent dynamics derived by the inferred future possibilities of a present experienced state arising from its past (the "perspective", intended both within a dynamic landscape and in the phenomenological[7,42] sense). We also find that the dynamics of the subcortex and cerebellum are affected in unconsciousness and speculate that emergent "apparent" dynamics related to experience will influence and be influenced by all processes related to in-formation.

## Materials and methods
### Cambridge anaesthesia dataset

*Participants.* Ethical approval was obtained from the Cambridgeshire 2 Regional Ethics committee. 18 out of 25 participants were used for the analyses due to incomplete data in the cortex and procedure failure. All participants were native English speakers (50% males) and healthy. Mean age was 33.3 (19–52). During scanning two senior anaesthetists were present. Electrocardiography and pulse oximetry were continuously measured. Blood pressure, heart rate and oxygen saturation were also monitored regularly.

*Anaesthetic protocol.* Propofol sedation was administered intravenously via "target Controlled infusion" with a Plasma Concentration mode. the Marsh pharmaco-kinetic model was used to control an Alaris PK infusion pump (Carefusion, Basingstoke, UK). This enables the anaesthesiologist to decide on a desired plasma

"target". This method permits the anaestheologist to regulate the infusion rates using patient characteristics as covariates. Three target plasma levels were used – no drug (awake control), 0.6 μg/ml (low sedation), 1.2 μg/ml (moderate sedation). In this study only the moderate sedation is used. fMRI data for this condition was acquired 20 minutes after termination of sedation, as a pharmacokinetic modelling software (TIVATrainer, www.eurosiva.org) predicted plasma concentration would reach zero after fifteen minutes. Before plasma target was changed, blood samples were obtained at the end of each titration period. The level of sedation was investigated verbally immediately before and after each scanning run. Three participants were not replying after the moderate condition but were easily awoken with loud commands.

10 minutes of plasma and effect-site propofol concentration equilibration was allowed before cognitive tests in the scanner were commenced (auditory and semantic decision tasks). During light sedation, mean (standard deviation) plasma propofol concentrations was 304.8 (141.1) mg/ml, During moderate sedation, 723.3 (320.5) mg/ml and 275.8 (75.42) mg/ml for recovery. Mean (SD) total propofol given was 210.15 (33.16) mg.

*Magnetic resonance imaging protocol.* A Trio Tim three tesla MRI machine (Erlangen, Germany), with a twelve-channel head coil was used to acquire 32 descending interleaved oblique axial slices with an interslice gap of 0.75 mm and an in-plane resolution of 3 mm. The field of view was 192 × 192, Repetition time and acquisition time was 2 seconds while the echo time was 30 ms and flip angle 78. A total of 5 min of resting state data was acquired for each condition. T1-weighted structural images with 1 mm resolution were obtained using an MPRAGE sequeunce with TR = 2250 ms, TI – 900 ms, TE = 2.99 ms flip angle = 9 degrees.

### Disorders of consciousness dataset (DOC)

*Patients.* MRI data for 23 DOC patients were collected Between January 2010 and July 2015 in the Wolfson Brain Imaging Center in Addenbrookes Cambridge, UK (mean time post-injury 15.75 for UWS and 16.9 for MCS). Data collection received approval from the National Research Ethics Service. These participants were selected out of a bigger dataset due to their relatively intact neuroanatomy. These patients were treated and scanned at the Wolfson Brain Imaging Center, Addenbrookes hospital (Cambridge, UK). Written Informed consent was obtained from an individual that had legal responsibility on making decisions on the patient's behalf. According to the diagnosis given by a physician, these participants were split into vegetative state/unresonsive wakefulness syndrome and minimally conscious groups (*n* = 12 for UWS and 11 for MCS). The mean CRS-r score was 8.3 (standard deviation 2.03), For the UWS group 7, (S.D. 1.41) and 9.75 (S.D. 1.54) For the MCS group. The mean age for the MCS group (39.18, S.D. 18.13); and for the UWS group was (40.16) S.D. 13.63. In the MCS group nine of the patients had a traumatic brain injury, one a cerebral bleed and one anoxia. In the UWS group the aetiology was defined as traumatic brain injury for 2 patients, one hypoxia, one edema, one having missing information and the remaining participants having the pathology caused by anoxia. In the MCS group 7 were male; whilst in the VS group 8 were male. Due to incomplete coverage of the cerebellum, one UWS participant was excluded from the cerebellar analyses.

*Magnetic resonance imaging protocol -DOC dataset.* A variable number of functional tasks, anatomical and diffusion MRI images were taken for the DOC participants. Only the resting-state data was used for this study. This was acquired for 10 minutes (300 volumes, TR = 2s) using a siemens TRIO 3T scanner. The functional images were acquired using an echo planar sequence. Parameters include: 3 × 3 × 3.75 mmm resolution, TR/TE = 2000 ms/30 ms, 78 degrees FA. Anatomical images T1-weighted images were acquired using a repetition time of 2300 ms, TE = 2.47 ms, 150 slices with a cubic resolution of 1 mm.

*Control Subjects DWI data.* Given that in the propofol dataset no DWI data was collected, we obtained data from control participants from another study looking at improvement in cognitive deficits in diffuse axonal injury patients with methylphenidate[95]. A total of 18 participants were included out of a total of 23. 5 participants were excluded due to incomplete data and equipment malfunction. The included participants (6 females) the mean age was 34 (SD = 10.7).

### London Ontario propofol (LON) dataset

*Participants dataset.* The second anaesthesia dataset was used to ascertain the reproducibility of results. This data was acquired at the Robarts Research Institute in London, Ontario (Canada) and was approved by the Western University Ethics board. 19 healthy (13 males; 18–40 years), right-handed, English speakers with no neurological conditions signed an informed-consent sheet and received monetary compensation for their time. The research ethics boards of Western University (Ontario, Canada) approved this study. Due to equipment malfunction or impairments with the anaesthetic procedure three participants were excluded (1 male). Thus, 16 participants were used for the reproducibility analysis. See reference[28] for the original study.

*Anaesthetic procedure LON dataset.* The scanning procedure was supervised by one anaesthetic nurse and two anaesthesiologists. Participants also completed an auditory target-detection and a verbal recall task as an additional assessment of level of awareness. An infra-red camera was also used to further assess level of wakefulness.

Propofol was administered intravenously using a Baxter AS50 (Singapore); stepwise increments were obtained via a computer-controlled infusion pump until all three assessors agreed that Ramsay 5 was reached (i.e., no responsiveness to verbal or visual incitements). When necessary further manual adjustments were made to reach target propofol concentrations. These targets were forecasted and maintained constant by a pharmacokinetic simulation software (TIVA trainer, European Society for Intravenous Anaesthesia, eurovisa.eu). The blood concentration levels following the Marsh 3-compartment model were also measured using this software. The initial propofol concentration target was 0.6 μg/ml, and step-wise increments of 0.3 μg/ml were applied after which Ramsay score was assessed. This process was repeated until participants ceased to answer verbally and where rousable only by physical stimulation. When this occurred data collection would begin. Oxygen titration was put in place to ensure SpO2 was above 96%. The mean plasma concentration was 2.68 (1.92–3.44) whilst the mean estimated effect site propofol concentration was 2.48 (1.82–3.14) μg/ml and propofol concentration. The mean total mass of propofol administered was 486.58 (1.92–3.44). Variability in pharmacokinetics and pharmacodynamics are characteristic in propofol administration. To ensure participant safety, scanner time was kept to a minimum although airway security via intubation could not be ensured. 8-minutes of resting state fMRI data was acquired.

*Magnetic resonance imaging protocol.* A 3-tesla Siemens Trio scanner was used to acquire 256 functional volumes (Echo-planar images [EPI]). Scanning parameters were: slices = 33, 25% inter-slice gap resolution 3 mm isotropic; TR = 2000 ms; TE = 30 ms; flip-angle = 75 degrees; matrix = 64 × 64. Order-of-acquisition was bottom-up interleaved. The anatomical high-resolution T1 weighted images (32-channel coil 1 mm isotropic voxels) were acquired using a 3D MPRAGE sequence with TA = 5 mins, TE = 4.25 ms, matrix = 240 × 256, 9 degrees FA.

*Acquisition of diffusion-weighted data.* The diffusion weighted MRI data was acquired over several years for the DOC data. During this period the diffusion weighted image acquisition scheme was changed. For the first 7 participants the data was obtained using an echo planar sequence (TR = 8300 ms, TE = 98 ms, matrix size = 96 × 96, 63 slices, slice thicknes = 2 mm, no gap, flip angle = 90°). In this acquisition the diffusion gradients were applied along 12 non-colinear directions with 5 b = 0 and 5 b-values that ranged from 340 to 1590 s/mm². The subsequent acquisition scheme was applied to the remaining 16 participants and the control cohort and used 63 directions with a b-value of 1000 s/mm². There is previous research on structural connectivity in DOC patients that use both of these DWI acquisitions[74,96]. Of the 7 participants that had data obtained via the 12 direction 5 b-values acquisition, 3 were diagnosed as MCS and 4 were diagnosed as UWS. To assess whether the different acquisition schemes may have affected results we ran a confirmatory analysis which excluded the participants with a 12-direction acquisition. The results reproduced and are presented in Supplementary Note 10.

*Diffusion weighted image preprocessing.* MRtrix3 tools was used to pre-process the DWI data (https://www.mrtrix.org/). The procedure involved denoising DWI data by investigating data redundancy using principal component analysis (MRtrix3 dwipreproc command, https://www.mrtrix.org/). Each subject's DWI data were aligned to the b0 image using FSL's eddy tool (MRtrix3 dwipreproc script). This same tool was used to correct for eddy current distortions. Subsequently diffusion gradient vectors were rotated to account for the subject motion estimated by the eddy FSL command. The b1 field inhomogeneities of DWI data were corrected using MRtrix3 dwibiascorrect command. Finally for each participant a binary mask was generated using the MRtrix3 dwi2mask command. When, under visual inspection, these masks were not deemed of sufficient quality, an alternative mask was generated using FSL's BET command and subsequently used for analysis.

The q-space diffeomorphic reconstruction (QSRD) of the DSI studio package[97] (www.dsi-studio.labsolver.org), was used to reconstruct DTI data from the preproccessed DWI images. This method calculated the distribution of orientations of the density of diffusing water in a normalised space. It does so by reconstructing the diffusion weighted images in native space and computing the quantitative anisotropy (QA) in each voxel. The QA values are subsequently used to normalise each subject's brain to a standard space (in this case MNI152) using the SPM nonlinear registration function. After having normalised the images to the standard space, spin density functions were reconstructed with a mean diffusion distance of 1.25 mm and having three fibre orientations per voxel.

*Diffusion tensor image data reconstruction and fibre tracking.* The deterministic tracking was performed on the reconstructed data using a modified FACT algorithm[98]. The following parameters were set as: angular cut-off = 55°, step-size = 1 mm, minimum length = 10 mm, maximum length = 400 mm, spin density function smoothing = 0, QA threshold as determined by the colony-stimulating factor DWI signal. The spatial termination of each streamline was automatically assessed. We applied the default anisotropy threshold (i.e., 0.6) to the SDF anisotropy values to create a white matter mask. This was used to discard streamlines

that ended prematurely in white matter. 1,000,000 streamlines were constructed for each individual.

Subsequently, the data were parcellated using parcellation schemes with different granularities. These parcellations are described in Supplementary Note 1.

*Preprocessing of functional and T1 images.* All functional images were preprocessed using the same in-house matlab script composed of SPM12 functions. After removing the first 5 scans, we performed slice-timing correction (reference slice = no. 17). After volumes were realigned to the mean functional image, these were normalised to an EPI-template using the function SPM's "old norm". This function was chosen as it was found, upon visual inspection, to produce the best results, similarly to previous work (Calhoun et al., 2017). Re-alignment parameters were produced and inserted as a covariate during time-series extraction. Participant specific grey matter, cerebral spinal fluid and white matter masks were also created using the segmentation function of SPM12. These were used to regress out spurious physiological signal in time series extraction. We visually inspected all normalised images with particular attention for the DOC dataset given the nature of this cohort.

*Time series extraction.* We performed Denoising using the SPM-based software CONN (17.f). We included movement parameters as a first-level covariate. We regressed out CSF and white matter signals from the timeseries using the first five principal components. To assess robustness of results we also ran an alternative analysis in which global signal regression was used (Supplementary Note 2). We used the ART quality-assurance/motion-artefact rejection toolbox (https://www. nitrc.org/projects/artifact_detect) to further clean the timeseries data. We applied linear detrending and a 0.008 to 0.09 Hz band pass filter. We also ran an alternative analysis in which a high pass filter was used (Supplementary Note 2). We used the nuisance variables obtained from the pre-processing of functional and T1 images (i.e., 5 principal components from white matter masks and movement parameters) to extract the timeseries. These timeseries were extracted in unsmoothed functional volumes to avoid inflated correlations in adjacent regions of interest (parcellation information presented in Supplementary Note 1; Table S1).

*Dynamic connectivity matrices definitions.* To obtain dynamic connectivity matrices for each participant the recommendations of Preti and colleagues were followed[99]. This included taking a 24-time point sliding window, and moving the window by one time point (2s = 1 TR) for the definition of each graph. Similarity to previous studies[37] this permitted to maximise the number of graphs available. A gaussian tempering was applied to de-weight the timepoints closer to the extremities of the sliding window.

Following this procedure resulted in 122 connectivity matrices for the anaesthesia dataset and 271 connectivity matrices for the DOC and 251 for the London Ontario anaesthesia dataset. For the meta-matrix construction, we took subsets of the data with higher numbers of timepoints in accordance with the dataset with least timepoints. For the relationship between structural and functional connectivity analysis we performed a confirmatory analysis using the minimum common denominator of timepoints (presented in Supplementary Note 10). We also applied an alternative dynamic functional connectivity method[100,101], namely instantaneous phase synchrony (Supplementary Note 2). This used a narrow band-pass filter (0.03:0.07 Hz) which is thought to satisfy Bedrosian's requirement for phase synchrony analysis[100,101] (the smaller the bandwidth, the more likely meaningful phases can be estimated; however see also[25]).

*The meta-matrix analyses.* The meta-matrix (MM) is a matrix displaying the similarity between all connectivity matrices. It is obtained by vectorising and correlating (Pearson's r) each connectivity matrix. Each column is organised linearly in time. We also reproduce results using alternative distance metrics to define the MM (similarity being the inverse of distance). These where the Manhattan distance and the cosine distance metrics[92,102] (Supplementary Note 2).

Visually, the MM is characterised by high similarities in proximal timepoints, which decay as a function of time (this is an over-simplification, see dMM Fig. 3c). We sought to explore whether a very simple synthetic model by which similarities linearly or exponentially decay over time could explain the intrinsic dynamics of certain conditions better. We named this the temporal decay of similarity model (TDSM) which was created using an in-house matlab script. This is a weighted model of decreasing similarity of connectivity matrices over time proximity (Fig. 1e in main text), and thus displays higher values along the diagonal (where cells represent similarities of proximal time-points). The TDSM essentially represents intrinsic dynamics that never return on themselves (monotonically lower values as a function of temporal distance), in which similarities decrease in a univocal and predictable manner (exponentially or linearly). Given the novelty of this method, we created three TDSMs; one with linearly decreasing similarity over time, one with slow exponential decreasing similarity over time, and one with rapid exponential decreasing similarity over time. The reason for the exponential models is that they fit the data better (mean r value = 0.70 SD = 0.04) than the linear model (mean r value = 0.53 SD = 0.05). The slowly decreasing exponential model was created with the following matlab code: exp(linspace(log(0.0001),log(1),n)/3), whilst the rapidly decreasing TDSM was created with

exp(linspace(log(0.0001),log(1),n)/1.5), n being the number of columns in the MM. The results were comparable between the linear and exponential TDSMs (Supplementary Note 3; Tables S2, S3).

To enable specific interpretation of the TDSM analyses, we then sought to analyse the properties of successive temporal transitions. This information is represented by the diagonals in the MM (excluding main diagonal which is always 1), which shows the similarity (i.e., distance) between two successive timepoints (Fig. 2a). Measuring the distance over time gives an approximation of the rate of change of intrinsic dynamics. We looked at the diagonals that are close to the main diagonal (Fig. 2b), indicating short term transitions. In the main text we show results for the first sub diagonal (thus each transition = 2s), but we repeated the analyses averaging across different sub-diagonals. Specifically, until the 6th (12s) and the 13th sub diagonal (26s), showing results are stable across temporal sampling of rate of change (Supplementary Note 4). On the resulting timeseries we calculated two measures of central tendency (mean and median; the latter reported in the main text). We also calculated two measures of the breadth of the distribution (standard deviation, and Shannon entropy, the former reported in the main text as Shannon entropy is unstable with low numbers of datapoints[55]. Finally two different measures of temporal complexity were calculated, effort to compress[55] and sample entropy (described below). Results reproduced robustly across the different measures (Supplementary Note 4; Tables S4, S5).

We then took a different approach and investigated the columns of the MM rather than the diagonals. The columns represent the similarity of one connectivity pattern to all others available (Fig. 3a, rather than transitions between connectivity patterns Fig. 2a). The aim of this analysis was to investigate the properties of the wider state space as represented via the intrinsic dynamics (MM). Thus, the similarity of one connectivity state in relation to all others indicates its position in the wider landscape of measured states (in the past and future; see Tononi et al., 2016; although their definition is probabilistic). To ensure the autocorrelation values (values close to the main diagonal) did not influence results, we removed this to create the distal meta-matrix (dMM; Fig. 3c). This off-diagonal MM represents the similarity between timepoints that are distant between each other. It is obtained by excluding the cells that are closest to the diagonal (which represent similarities between connectivity matrices that are closer in time). An arbitrary number of diagonal cells had to be chosen for exclusion. In this case we chose 13 as is it half the number (+1) of timepoints included in each sliding window. However, we also repeated the analyses using 24 cells for exclusion, thus ensuring the column of interest had no datapoints in common with the connectivity matrices it was compared to. On the distal timepoint MM (dMM), we calculated measures of central tendency (mean and median), distribution variation (standard deviation and Shannon entropy) and sequential complexity measures (sample entropy and effort to compress, described below; Tables S6, S7). We repeated the analysis by vectorizing the whole dMM instead of calculating the measures in a column specific manner (i.e., position in state space of a connectivity pattern), and found results reproduced, albeit with slightly lower effect sizes (Supplementary Note 5). In the main text, results for the lower granularity parcellation (see Supplementary Note 1; 126 regions; whole brain) for the Cambridge anaesthesia dataset ordered with the DOC dataset are presented. See Supplementary Notes for reproducibility of the independent predictive power of these measures (Tables S8, S9) and of different subsystems (Tables S10–S12).

*DTI to FC dynamic analysis.* This analysis was created to test the hypothesis that functional connectivity states had more freedom to vary in relation to the underlying structural connectivity. The functional and structural data was parcellated with the same parcellation scheme and the functional matrices were thresholded proportionally (taking a top percentage of connections) to match the number of non-empty cells of the structural connectivity matrices on an individual-by-individual basis. A similar approach to the MM was used by which the similarity of all connectivity states (organised linearly in time) to the structural connectivity matrix was calculated (using Pearson's R). This yielded a vector of numbers indicating the similarity in time of the functional connectivity state to the structural connectivity. On this vector both effort-to-compress and sample entropy were calculated, thus obtaining one number for each participant that was inserted into the ordinal logistic regression (described below) as a predictor. Given that there were differing numbers of available timepoints between the DOC and control data we reduced the DOC data timepoints to the lowest common denominator. Results reproduced across these analysis permutations (Supplementary Note 10, Tables S13, S14). Furthermore, we reproduced the cortex, subcortex and cerebellar results using alternative distance metrics to evaluate the structural functional relationship. Similarly to the reproductions of the MM results, we used Manhattan and Cosine distance. Whilst the Cosine distance reproduced results univocally, the Manhattan distance only reproduced subcortical results (Supplementary Note 11; Tables S15–S18).

*Entropy measures.* Entropy can be considered a measure of the rate of information produced in dynamical systems. Sample entropy was specifically developed to ameliorate the problems typically encountered in biological time-series (i.e., low number of time-points and noise[54]). Sample entropy is conceptually derived from Kolmogorov Complexity[103]. This states that if a complex system cannot be quickly and easily summarised, then it is complex. Such complexity measures have been

shown to reproduce across datasets and correlate to other complexity measures[104]. Implementations of such notions of complexity has also been shown to correlate with the meaningfulness of naturalistic stimuli; in-scanner behaviour, to be reproducible across fMRI sessions and be robust to different parameters[105–110].

Sample entropy works by taking two segments of timeseries of different lengths and contrasts to what extent each of these timeseries explain the rest of the data (via a distance measure, in this case Chebyshev). Sample entropy is calculated as the ratio of the explanatory power (calculated via distance metric) of the smaller (denoted A) and the larger segment (B) of data.

$$SampEn = -\log\frac{A}{B}$$

Thus, higher values indicate decreased self-similarity, as the smaller segment explains the data better and would indicate a fine grain description is more appropriate, therefore indicating the signal is more complex.

We chose sequence lengths (A & B) of 2 and 1 as this produced robust results and has been used in the past[105]. As suggested by the original paper[54] we chose a tolerance of 0.2 multiplied by the standard deviation of the time-series.

Given the novel approach used in this paper, we sought to confirm results with an alternative entropy measure. Also derived from Kolmogorov complexity, Effort-To-compress (ETC) relies on a lossless compression algorithm via Non-Sequential Recursive Pair Substitution[55]. This algorithm has been shown to be an improvement in efficiency compared to Shannon entropy and the Lempel-ziv compression algorithm when applied to short and noisy sequences of data.

ETC works by iterating over the given sequence and substituting the most frequently occurring pattern of a pre-established length with a new symbol. For example, with the binary input of "00101101" and length of two, the algorithm would substitute the pair "01" with a new symbol (i.e., 2) and that pair is the most frequently occurring. Thus, the output string would be 02212. Given parity between occurrence frequency of each pair, the first pair is substituted with the new symbol "3", thus giving 3212. This process is repeated until the output becomes constant or it is reduced to one number. Thus 3212-> 412->52->6. In this example there are 5 iterations the algorithm needs to perform, thus indicating the degree to which that sequence can be easily described, and therefore it's (Kolmogorov) complexity. Given that the highest possible output of ETC is the length of the original series minus 1, the measure can be normalised thusly:

$$ETC_{norm} = \frac{ETC}{L-1}$$

In the case of real numbers, as is the one in this paper, the numeric sequence is translated to a symbolic sequence by binning the data. We chose to use 10 bins as higher number of bins seem to produce more stable results in shorter sequences[55].

These measures (effort to compress and sample entropy) however consider sequential information and show similar results. Shannon entropy on the other hand is sensitive to the probability of values independently from sequential information. Hence, we used Shannon entropy (as implemented in matlab) to confirm the standard deviation results (variation in proximal timeseries).

*Inferential analyses: ordinal logistic regression.* To assess the hypothesis that the dynamical complexity of connectivity states augmented with increasing levels of awareness, ordinal logistic regressions were performed using the MASS R toolbox (polr function, https://www.rdocumentation.org/packages/MASS/versions/7.3-53/topics/polr). This is a regression model with ordinal categorical dependent variables and continuous independent variables. This is derived from the logistic regression and ideally suited to this study for the little assumptions underlying it. Nonetheless, multicollinearity was assessed when multiple predictor variables were included and the proportional odds assumption was tested using Brants test (using package 'brant'); and where this failed (in RSFC to DTI analysis) by comparing the model with parallel intercepts constrained and the same model without constrained intercepts. The proportional odds assumption entails the model coefficients have a proportional effect on each group; i.e., "the slope" estimated between each condition (outcome variable) is the same or proportional. Whilst the TDSM and structure function analyses were one sided (given the strong prediction that complexity would increase in consciousness); the proximal and distal measures were running as two sided given the absence of a strong hypothesis, specifically in regards to measures of central tendency and distribution breadth. Nonetheless, in the one-sided analyses, *p* values were always below 0.025. To increase interpretability of Odds ratio values reported in the main text, these were calculated with inverse ordering of conditions (i.e., unresponsive wakefulness syndrome > minimally conscious state > sedation > control awake) in analyses that predicted increases with levels of awareness (complexity of dMM and relationship between structural and dynamic functional connectivity).

## Data availability

The London Ontario Dataset is available (https://openneuro.org/datasets/ds003171). Due to the clinical nature of the data, this will be made available upon reasonable request to the corresponding author. All toolboxes used in this study are freely available and cited appropriately in the text.

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

## Acknowledgements

Wellcome Trust: Clinical Research Training Fellowship 083660/Z/07/Z (RA) UK Medical Research Council (U.1055.01.002.00001.01) (A.M.O., J.D.P.) The James S. McDonnell Foundation (A.M.O., J.D.P.) The Canada Excellence Research Chairs program (215063) (A.M.O.) The Canadian Institute for Advanced Research (RCZB/072 RG93193) (A.M.O., D.K.M., and E.A.S.) Cambridge Biomedical Research Centre and NIHR Senior Investigator Awards (J.D.P. and D.K.M.) The British Oxygen Professorship of the Royal College of Anaesthetists (D.K.M.) The Evelyn Trust, Cambridge and the EoE CLAHRC fellowship (J.A.) The L'Oreal-Unesco for Women in Science Excellence Research Fellowship (L.N.) The Stephen Erskine Fellowship, Queens' College, University of Cambridge (E.A.S.) Computing infrastructure at the Wolfson Brain Imaging Centre (WBIC-HPHI) the NIHR Brain Injury Healthcare Technology Co-operative based at Cambridge University Hospitals NHS Foundation Trust Cambridge Trust (P.C.) MRC research infrastructure award (MR/M009041/1).

## Author contributions

Conceptualization: P.C., E.A.S. Methodology: P.C., E.A.S. Supervision: E.A.S., D.K.M. Writing- Original Draft: P.C., E.A.S. Writing – review and editing: E.A.S., L.N., A.M.O., D.K.M., J.A., P.F., R.A., G.B.W., E.A.S., J.D.P. Data collection: A.M.O., D.K.M., J.A., P.F., R.A., G.B.W., E.A.S., L.N., J.D.P. We would like to thank Victoria Lupson and the staff in the Wolfson Brain Imaging Centre (WBIC) at Addenbrooke's Hospital for their assistance in scanning. We would also like to thank Jeffery Yoshimi for his invaluable insights and comments on this paper and the topics pertaining to it.

## Competing interests

The authors declare no competing interests.
