## [Peer Review File · Communications Biology]

Reviewers' comments:

Reviewer #1 (Remarks to the Author):

This is an excellent paper by one of the major groups on the disorders of consciousness. They start out by a novel way of dynamic functional connectivity and its dynamic landscape in order to link it to the phenomenology of consciousness, namely the flow or stream of consciousness in individual subjects. They show that this dynamic repertoire is decreased in subjects who lost consciousness in a variety of different states. This is an outstanding paper that innovates a novel methodology to link neural dynamic and phenomenology in consciousness. Some small remarks to this excellent paper.

- Tagliazuchi et al. (2013) argue that the temporal continuity on the neural level as measured by scale-free activity may be related to the stream of consciousness...these and other approaches should be discussed and compared with the own..

- The introduction opens with a general statement about Descartes...I am not really sure whether fits here..

- Their analyses technique sounds like an autocorrelation window/similarity of functional connectivity states across time; this is reflected in the name of proximal timepoint similarity model (pTSM)...

- ...how do their results stand in relation to the intrinsic neural timescales which are known to be altered in disorders of consciousness (Zilio 2021, Neuroimage); this should be made clear and discussed...

- The ETC Index is a version of the LZC, as I understand it? how do you make sure that it is valid given the BOLD signal and its temporal sluggishness..some are sceptical about applying time compression indices like LZC to fMRI data...you may need to justify and validate that to convince those sceptics....

- They investigate different measures, entropy for structural and functional c, co-complexity for distal C, and similarity/autocorrelation for proximal...how are all these measures related to each other? It would have been easier if they apply the same measure or somehow correlate them with each other...this would make the interpretation easier...

- How did they deal with the lesion issue in the UWS and MCS subjects? This is especially relevant for their region based analyses...

- The link of neural dynamic and the stream of consciousness has also been described by earlier work by the group around Northoff in some papers and also in the first chapters of *Unlocking the brain Vol II*. I recall another paper 2010 in *J Cognitive neuroscience* making the connection between the stream of consciousness/phenomenology and the analyses of neural data...

- How about the global signal? A recent study by Tanabe et al. (2020) showed that the level of the global signal is parametrically related to the level/state of consciousness in various altered states. How do the authors deal with that? Ideally I would like to see their data with and without global signal regression....

- I would like to see the complexity applied to proximal series and the similarity to distal..but may be I overread that...

- Introduction: line 80 – this is the protention-retention approach by Husserl...this was addressed by others like Northoff and the other studies of Tagliazuchi, Northoff, etc mentioned

- My feeling is that there is a phenomenological overinterpretation: they measure the neural continuity with their autocorrelation-like analysis of functional connectivity pattern....but does that imply showing that the subjects really experience or do not experience a flow or stream of consciousness? That remains an inference and not implied by the data...Hence, there is no real convergence of dynamic and phenomenological features...as they do not have any subjective data...so the main issue is a new very well done analysis of functional connectivity patterns...

- The temporo-spatial theory of consciousness (TTC) should be discussed here....it fits much better the results than the IIT....also the paper by Lamme and Northoff 2020 may be important to cite as it compares different theories of consciousness...

Reviewer #2 (Remarks to the Author):

In this article, the authors demonstrate the link between brain dynamic functional connectivity with theories of consciousness, providing solid evidence that the evolution of functional connectivity patterns over time is intrinsically related to consciousness level. The study is presented in a very clear way, targeting a broad audience in neuroscience, linking recent developments in neuroimaging and psychology to address fundamental aspects of brain function.

The authors propose rigorously validated measures that demonstrate robustness across datasets, parcellation schemes and methodological parameters (i.e., window length). Further, the results are shown not to be exclusive to the cortex and to replicate in subcortical structures and the cerebellum.

The technical implementation demonstrates expertise, keeping only the necessary degree of complexity inherent to the metrics employed, while maintaining a sufficient degree of interpretability.

However, some issues should be addressed to improve the quality of the manuscript before considering publication in *Communications Biology*.

General comments:

1 - 'We provide evidence for the common intuition that increasing levels of awareness are characterised by a rich "stream of consciousness" or flow of mental states (measured indirectly via the functional network variation intrinsic to the subject), that re-organise rapidly, are specific (informative) across longer periods of time and have more freedom from their structural underpinnings.'

Although the results presented herein reveal a clear relationship between the metrics employed and the levels of consciousness, the relationship with the 'stream of consciousness' can only be speculated, since it is unclear if and how the different connectivity patterns actually reflect mental states, so the authors could maybe rephrase to temper the claims.

Most importantly, the authors should revise what is understood by 'rapid' (used several times throughout the text), since the results presented reflect only a stream occurring on a ultra-slow timescale (i.e. first 0.008 to 0.09 Hz band-pass filter + sliding 24s-window correlations). Indeed, it would really reinforce the validity of this work, and perhaps even improve the sensitivity of the measures, if the authors show that the results survive on faster evolutions of fMRI FC, which can still be detected in the fMRI signal.

In more detail, several works have shown that using the instantaneous phase of the BOLD signal allows the calculation of FC at the instantaneous level (Glerean et al, 2012), which significantly improves the results in comparative studies of time-versus-time dynamic FC matrices (i.e., Cabral et al., 2017). Basically, this allows calculating the Meta-Matrix without the smoothing induced by the sliding window, in order to verify how fast the 'stream of consciousness' can be detected. If this is not possible, at least the authors should mention this direction for future studies.

2 - 'We applied linear detrending and a 0.008 to 0.09 Hz band pass filter to eliminate scanner-related noise.'

This filter eliminates more than just scanner related noise, and removes also other physiological components, so the sentence should be corrected. If possible and feasible, it would be valuable if the authors could explore whether the results presented herein survive (or even improve), when considering a broader frequency range? For instance, recent studies have demonstrated that including physiological signal may have important contributions to resting-state activity, and therefore it would be important to evaluate if extending the frequency range improves the biomarking potential of the measures (Chen et al., 2020; Vohryzek et al., 2020). If this is not possible, at least the authors should

mention it in the discussion.

3 - 'We calculated complexity measures (ETC and sample entropy) on each column of the distal meta-matrix and averaged across these.'

Why didn't the authors compute the complexity measures directly in the vectorized upper triangular part of the dMM matrix?

Small corrections:

Figures:

Figure 1

Panel C – add colorbar;

panel D – add axis labels;

panel F Increase font sizes.

Legend for panel F is missing.

Figure 3 – A,B,C is in the legend but not in the figure.

Figure 4 – Missing title.

Also, it would be important to explain that the structural connectivity is obtained from applying tractography algorithms to DTI images.

Please revise the text in the following lines:

128-129 revise sentence

155 widow

160 represent

304 suggest

305 ; and

363 references should be cited as numbers

Additional references:

Glerean, E., Salmi, J., Lahnakoski, J. M., Jääskeläinen, I. P., & Sams, M. (2012). Functional magnetic resonance imaging phase synchronization as a measure of dynamic functional connectivity. *Brain connectivity*, 2(2), 91-101.

Cabral, J., Vidaurre, D., Marques, P., Magalhães, R., Silva Moreira, P., Miguel Soares, J., Deco, G., Sousa, N., Kringelbach, M.L., 2017. Cognitive performance in healthy older adults relates to spontaneous switching between states of functional connectivity during rest. *Sci Rep* 7, 5135. <https://doi.org/10.1038/s41598-017-05425-7>

Chen, J.E., Lewis, L.D., Chang, C., Tian, Q., Fultz, N.E., Ohringer, N.A., Rosen, B.R., Polimeni, J.R., 2020. Resting-state "physiological networks." *NeuroImage* 213, 116707. <https://doi.org/10.1016/j.neuroimage.2020.116707>

Vohryzek, J., Deco, G., Cessac, B., Kringelbach, M.L., Cabral, J., 2020. Ghost Attractors in Spontaneous Brain Activity: Recurrent Excursions Into Functionally-Relevant BOLD Phase-Locking States. *Front. Syst. Neurosci.* 14, 20. <https://doi.org/10.3389/fnsys.2020.00020>

Reviewer #3 (Remarks to the Author):

This paper proposed a method to investigate the temporal complexity of functional connectivity patterns, and showed that it scales with the level of awareness, and also demonstrated that the cortex, cerebellum and subcortex all display consciousness-relevant dynamics. Overall, I think this is a well-conducted study, which contributes to the field by providing a new method and an alternative perspective to study dynamic functional connectivity. However, I think the manuscript could be improved in the following aspects.

1. the similarity between connectivity matrices was assessed by Pearson's correlation. What is the rationale for this choice instead of other measures (e.g., a distance measure)? As the similarity is the basis to quantify complexity in this study, I'd suggest testing the reproducibility using alternative similarity methods.
2. Could you provide and discuss the pairwise comparison (as did in structural-functional similarity section Line 296-303) results for Figure 1-3? In most of the pTSM cases and some dMM cases, it looks like no obvious difference between consciousness and sedation states, despite of significant odd ratios across all states.
3. Comparing the results of whole brain, cortex, subcortex, and cerebellum, I wonder whether odd ratios to some extent can imply the performance of complexity in prediction of conscious state? For example, in Figure 4, OR is higher for cortex than subcortex and cerebellum; but how to understand the OR is the lowest among the four?
4. The results suggest that cortex, cerebellum and subcortex all display consciousness-relevant dynamics. I think it would be helpful if the authors could further explore (use a statistical method?) whether subcortex and cerebellum could provide additional information in the prediction of conscious state beyond cortex
5. Figure 1. The matrices in C and E should be represented in the same size if they are. Please indicate which model was used in E
6. Figure 2. please specify the measures and parameters used in the figure (12 or 24 timepoints, ETC or sample entropy), instead of describing the method in general
7. Figure 4. Kolmogorov complexity measures?

We would like to kindly thank the reviewers for their constructive comments which we address below individually. The reviewers' comments are in black, our responses are in red text, whilst the revised text from the article is indented.

Because of the substantial number of new and confirmatory results we produced in response to the reviewer's comments, we uploaded some of them onto github and we have made them accessible through links in the supplementary materials (more details below).

Reviewers' comments:

Reviewer #1 (Remarks to the Author):

This is an excellent paper by one of the major groups on the disorders of consciousness. They start out by a novel way of dynamic functional connectivity and its dynamic landscape in order to link it to the phenomenology of consciousness, namely the flow or stream of consciousness in individual subjects. They show that this dynamic repertoire is decreased in subjects who lost consciousness in a variety of different states. This is an outstanding paper that innovates a novel methodology to link neural dynamic and phenomenology in consciousness. Some small remarks to this excellent paper.

Firstly, we would like to thank the reviewer for the positive comments. In particular we appreciate the reviewer's willingness to engage with the ideas presented in our work and their very thought-provoking comments.

- Tagliazucchi et al. (2013) argue that the temporal continuity on the neural level as measured by scale-free activity may be related to the stream of consciousness...these and other approaches should be discussed and compared with the own..

This is a very interesting paper showing that the long-range temporal autocorrelation or memory in the timeseries is reduced in the most complex regions of the brain (according to the definition in Margulies et al., 2016). This temporal autocorrelation, measured via the Hurst exponent, cannot be detected with simple measures of variance. It should be noted that this example of Tagliazucchi's work focuses on activity rather than connectivity patterns. Nonetheless, this is a very relevant reference that has been added, in the context of broader implications the temporal-spatial organisation of the brain.

The reviewer's comments (above and below) motivated us to add the following section in the discussion.

Lines: 404-410

The intrinsic temporal organisation of neural activity has been thought to provide a fundamental scaffold on which unified experience may emerge (Huang et al., 2021; Northoff & Huang, 2017; Tagliazucchi et al., 2013; Zilio et al., 2021). Previous studies have in fact showed a disruption of intrinsic temporal structure (e.g., self-similarity) of BOLD and electroencephalogram signals in different states of unconsciousness (Liu, Ward, Binder, Li, & Hudetz, 2014; Luppi et al., 2019; Tagliazucchi et al., 2013; J. Zhang et al., 2018; Zilio et al., 2021). Here we show that in unconsciousness, temporal organisation is affected also at the network level in specific ways (i.e., with slower average speed but relatively more acceleration and deceleration, with more predictable transitions and a less complex state space).

- The introduction opens with a general statement about Descartes...I am not really sure whether fits here..

We understand the reviewers concerns about this general statement. This was primarily a reference to the so called "hard problem of consciousness". How and why is it possible for subjective qualitative experience to arise out of matter. We have modified that phrase as follows.

Lines: 43-44

neuroscientific endeavour encounters fundamental questions: how do the material and the experiential correspond?

- Their analyses technique sounds like an autocorrelation window/similarity of functional connectivity states across time; this is reflected in the name of proximal timepoint similarity model (pTSM)...

We would like to thank the reviewer for their insightful comment.

We agree that autocorrelation is relevant to our analyses. We feel we have also gained a more detailed understanding of this analysis, also thanks to the reviewer's comments. Although not a typical way of measuring autocorrelation (we look at it in terms of connectivity patterns when autocorrelation is typically discussed in terms of a bidimensional signal (e.g., Tagliazucchi et al., 2013)), it is possible that the Temporal Similarity Decay Model (pTSM in previous version) does contribute something further than typical autocorrelation measures, in that we actually try to predict all of the individual's intrinsic dynamics (beyond the proximal, predicting the whole meta-matrix structure) using diminishing similarities. Thus, this model would represent dynamics that predictably diminish in similarity over time and never return to a previous state (in contrast to metastability [Tognoli & Kelso, 2014]; see also William James (James, 1890)).

Thanks to the reviewer's comments, we investigate the complexity of autocorrelation specifically (see figure 2 in the main text (...), without the more distal (e.g., >26 seconds) dynamics). In view of these considerations, we have changed the interpretation of the TDSM (previously proximal time similarity model).

Lines: 140-145

We sought to investigate the predictability of the intrinsic dynamics of individuals in different states of consciousness. To assess this, we constructed a simple synthetic model (Fig. 1E) in which the similarity between different connectivity patterns decayed monotonically as a function of time. Thus, in this idealised model, timepoints that were further away always displayed less similarity than closer timepoints, effectively modelling a simple dynamic trajectory in state space (e.g., Fig. 1D) that never returns on itself.

[...]

Lines: 151-164

As expected, unconscious conditions were more similar with the temporal similarity decay model (TSDM) than the more conscious states (Odds Ratio (OR):3.17 p=0.0001 C.I. (2.5%:97.5%)=1.78:6.22; Fig. 1F). Remarkably, this effect was robust across different TDSM models (linear or exponential decays), parcellations, data and pre-processing pipelines (except when high pass filter was used; see methods and S3). This may suggest that when a subject is unconscious, their immediate past and future states are more predictably similar to the present state, indicating sluggishness in connectivity state reconfiguration. Alternatively, this may indicate that there is little similarity of states over longer periods of time. Of note is that the TDSM model explained the (MM) intrinsic dynamics of certain UWS patients particularly well (Fig., 1F). Thus, gradual, non-recursive transitions in connectivity patterns poorly described conscious individuals which indicates these may be characterised by short term unpredictable FC reconfiguration, complexity of distal state exploration and potentially intrinsic self-organisation. To further tease apart these possible effects, below we explore the properties of short (2s) and long term (>26s) intrinsic dynamics.

The reviewer is perfectly right to point out that proximal transitions of network configurations relate to autocorrelational information. In view of this, we have explicated this relationship in several points of the manuscript

In the results, introducing the new proximal analyses:

Lines: 192-195

We investigated to what extent shifts in sequential connectivity states that are temporally close to each other may be predictive of levels of awareness (Fig. 2A). This autocorrelational information is represented in the MM by the values close to the main diagonal (Fig. 2B, in green) which are characterised by high similarity.

In the discussion

Lines: 409-413

The temporo-spatial theory of consciousness (Northoff et al., 2017) postulates that the intrinsic temporal-spatial organisation of the brain is fundamental to the emergence of consciousness. In accordance to this theory's predictions, we show that consciousness is associated with specific autocorrelation properties (Fig 2C, 2D, and 2E) and a more complex dynamic repertoire (Fig. 3E) which would theoretically support subjective feelings of continuity and fluctuations of states of consciousness over time (Northoff, Wainio-Theberge, & Evers, 2020).

- ...how do their results stand in relation to the intrinsic neural timescales which are known to be altered in disorders of consciousness (Zilio 2021, Neuroimage); this should be made clear and discussed...

This is a very interesting paper showing that several intrinsic temporal properties are affected in conditions with altered sensory processing (which could also be called unconsciousness e.g., Unresponsive wakefulness syndrome vs Locked in Syndrome). This is also very much in line with our results and can be connected to the paper by Tagliazucchi (2013) suggested above by the reviewer. These and other studies (e.g., Zhang 2018; 2021) have indeed shown a change in the intrinsic temporal structure of neural processes during unconsciousness, which according to theory (temporo-spatial theory of consciousness (Northoff & Huang, 2017) may be a fundamental mechanism sustaining consciousness. After the reviewer's comments, we feel that one of the contributions of our study is showing that the intrinsic temporal organisation of the brain is affected in unconsciousness at the dynamic large-scale network level. The use of the present paradigm permits to measure such an intrinsic organisation (e.g., self-similarity in time/autocorrelation) of multidimensional connectivity patterns.

Lines: 404-406

Previous studies have in fact showed a disruption of intrinsic temporal structure (e.g., self-similarity) of BOLD and electroencephalogram signals in different states of unconsciousness (Liu, Ward, Binder, Li, & Hudetz, 2014; Luppi et al., 2019; Tagliazucchi et al., 2013; Zhang et al., 2018; Zilio et al., 2021).

- The ETC Index is a version of the LZC, as I Understand it? how do you make sure that it is valid given

the BOLD signal and its temporal sluggishness..some are scepticla about applying time compression indices like LZC to fMRI data...you may beed to justify and validate that to convince those sceptics....

We understand that LZC and ETC are both based on the same notion, sometimes termed Kolmogorov or algorithmic complexity (Kolmogorov, 1965; Mitchell, 2011). The principle behind this is that if information can be summarised efficiently, then it is not complex. If a summary of the information is not much shorter than the original information, then it is complex. However, ETC and LZC are implemented differently. In their proposal of ETC, the authors (Nagaraj, Balasubramanian, & Dey, 2013) make the point that ETC is more robust for short and noisy time signatures compared to LZC, therefore perhaps making it more suitable than LZC in the present case. Nonetheless, it is worth pointing out that in a counterbalanced repeated measure designed study (Boly et al., 2015), LZC of BOLD signal correlated with the meaningfulness of naturalistic stimuli, thus suggesting that LZC may in fact be a meaningful metric when applied to BOLD signals. Additionally, a study by Varley and colleagues (2019) showed that LZC of BOLD and vectorised connectivity patterns correlated with other complexity measures (including Hurst exponent used by Tagliazucchi et al., 2013) consistently across datasets.

In this study we reproduced all ETC results using sample entropy (and vice versa, presented in supplementary materials). Sample entropy, is based on the same principle of Kolmogorov complexity that underlies LZC and ETC, although implemented differently (explained in methods section).

This is a measure that was specifically devised to measure the complexity of dynamic (noisy) physiological signals (Delgado-Bonal & Marshak, 2019; Lake, Richman, Pamela Griffin, & Randall Moorman, 2002; Richman & Moorman, 2000) and is shown to have marked advantages from its predecessor (i.e., approximate entropy). Sample entropy has been specifically applied to the BOLD signal, an implementation of which can be found in this toolbox (Wang, Li, Childress, & Detre, 2014; <https://cfn.upenn.edu/zewang/BENTbx.php>) which has been used to find consciousness relevant results (Luppi et al., 2019).

Furthermore, this measure has been shown to correlate with fluid intelligence and in-scanner behaviour, to be reproducible across different fMRI sessions and to maintain robustness across different parameters (Omidvarnia et al., 2021; Pedersen, Omidvarnia, Walz, Zalesky, & Jackson, 2017; Wang et al., 2014; S. Zhang, Rogers, Morgan, & Chang, 2020).

In our work, we show that ETC and Sample Entropy results reproduce well across the different datasets used and across different methodological contingencies (e.g., how many proximal timepoints to remove in distal meta matrix, size of ROIs forming networks; including additional pipelines proposed by the reviewers [e.g., high pass filter, alternative distance metrics]). Using such algorithmic/Kolmogorov complexity measures has the advantage that they are sensitive to the temporal sequence of the data (by measuring predictability in adjacent data), information which would be lost if Shannon entropy were used (as shown in figure 2D and 2E in the paper).

We hope these considerations may ease scepticism in regards to the use of these measures. In order to make these considerations more explicit we have added the following text when we describe the sample entropy and ETC measures in the method section.

Lines: 826-830

Such complexity measures have been shown to reproduce across datasets and correlate to other complexity measures (Varley et al., 2020). Implementations of such notions of complexity has also been shown to correlate with the meaningfulness of naturalistic stimuli; in-scanner behaviour, to be reproducible across fMRI sessions and be robust to different parameters (Boly et al., 2015; Coppola et al., 2022; Omidvarnia et al., 2021; Pedersen et al., 2017; Wang et al., 2014; S. Zhang et al., 2020)

- They investigate different measures, entropy for structural and functional c, coimplexity for distal C, and similarity/autocorrelation for proximal...how are all these measures related to each other? It would have been easier if they apply the same measure or somehow correlate them with each other...this would make the interpretation easier...

This is an extremely pertinent point. Following the reviewer's comment, we investigated the relationship between all measures for the meta-matrix analyses (comparison with structural and functional results not possible due to use of different participants).

Please see below the correlation matrix between these measures (some of these measures were not present in the first draft of the manuscript and will be explained in full below). We reproduced the correlation matrices across different granularity parcellations and using the alternative anaesthesia dataset (see supplementary material 7).

In accordance to the reviewer's concern, we also inserted the most predictive temporal and distal measures within the same ordinal logistic regression to investigate whether these had independent predictive power of the levels of consciousness. However, these results did not reproduce very well across analyses (we have inserted the relevant section from the supplementary materials for the reviewer's benefit).

E Spearman correlation between proximal and dMM measures

Figure 3E shows the shared variance between the distal and proximal measures (spearman correlation; $p < 0.001$). dMM=distal meta-matrix, ETC = Effort-to-compress; SampEn = Sample entropy; SD = standard deviation; prox = of proximal temporal transitions.

Lines: 296-309

Relationship between distal and proximal measures.

We found that with consciousness, proximal transitions tend to be less predictable, quicker, but more constant. Furthermore, the wider dynamic state space (approximated by the dMM), seems to be more complex (“less compressible”) in consciousness. We investigated to what extent these measures held unique variance. We found that, all variables were correlated to each other ($p < 0.001$), although there seemed to be evidence the different measures displayed some independent variance (Fig. 3E, averaged explained variance 22%), specifically between proximal (Fig 2C, 2D, and 2E) and distal measures (fig 3D; see S7 for reproducibility). We then inserted the variables with the highest effect sizes for the proximal and distal measures into the same ordinal logistic regression as covariates (namely sample entropy of proximal transitions and compressibility of dMM). We found that, there was evidence each of these had independent predictive power (dMM compressibility OR= 3.82, $p=0.0004$; Proximal SD OR=1.93, $p=0.02$), although this did not reproduce unequivocally across different control analyses (S7 for details). We therefore cannot conclude whether proximal and distal measures have unique predictive power.

Please see the following section from supplementary material 7 for details on levels of reproducibility when proximal and distal measures were inserted into the same ordinal logistic regression.

Reproducibility across alternative methods of unique predictive variance of proximal and distal measures when inserted as covariates in the same ordinal logistic regression.

Whilst with global signal regression and high pass filter, only the proximal measures were significant; in the alternative anaesthesia dataset (with aCompCorr and bandpass filter between 0.008 and 0.09; shown in table S9) and with the use of instantaneous phase synchrony (IPS), the majority of dMM analyses were significant (except lower granularity for IPS, $p=0.06$). When the Manhattan distance measure was used, both measures were significant, only in the higher granularity parcellation, whilst only the proximal measure was significant in the lower granularity parcellation. With the cosine distance both measures were significant. Full results can be found at https://github.com/Peter6789/Meta-Matrix_reproduction/tree/main/Proximal_and_Distal_in_same_OLR. This indicates that the degree of independent predictive power of each measure varies as a function of methodological contingencies. Thus, given the relative symmetry between proximal and distal results, we are not able to conclude whether either one has more predictive power, or whether they have independent explanatory power.

- How did they deal with the lesion issue in the UWS and MCS subjects? This is especially relevant for their region based analyses...

This is also a very important point and we thank the reviewer for raising it. This is effectively an opportunity DOC sample. We selected these patients out of a large cohort ($n=71$) due to the relatively intact anatomy which was extensively visually assessed. This may have biased the sample in terms of extrapolating to the entire DOC population. However, including more participants would have biased the analyses in other ways, as the reviewer suggests. Additionally, we would argue that the approach set out in this work does in part obviate the problem of comparability between different conditions (such as traumatic brain injury, ischemic injury patients and healthy controls). This is because we do not compare the spatial connectivity per say, which would be highly influenced by lesions. We compare how the spatial connectivity patterns varied *within an individual*. By using the subject as their own “baseline” (by computing differences given a specific temporal distance from an origin), the meta-matrix abstracts the spatial patterns to the temporal domain in a subject-specific manner. We argue that this abstract temporal space enables a relatively more uncontroversial comparison between

different subjects and conditions. We compare the “distance travelled over time” in a subject rather than the connectivity patterns of specific regions. The reproducibility of results using an alternative anaesthesia dataset (collected in a different site) hopefully should further mitigate concerns that results were driven by specific lesions in patients.

Nonetheless, these patients have unusual brains, in that most of them are injured and disproportionately atrophied and it should be noted that in terms of preprocessing we tried several normalisation procedures and found that normalising the fMRI images to an EPI template directly produced the best results (see also Calhoun et al., 2017). The resulting normalised images were extensively visually assessed.

We have added this phrase in relation to this (note, “this function” refers to the direct normalisation of functional images to the EPI template)

Lines: 708-709

This function was chosen as it was found, upon visual inspection, to produce the best results, similarly to previous work (Calhoun et al., 2017).

Lines: 713-714

We visually inspected all normalised images with particular attention for the DOC dataset given the nature of this cohort.

- The link of neural dynamic and the stream of consciousness has also been described by earlier work by the group around Northoff in some papers and also in the first chapters of *Unlocking the brain Vol II*. I recall another paper 2010 in *J Cognitive neuroscience* making the connection between the stream of consciousness/phenomenology and the analyses of neural data...

We would like to thank the reviewer for pointing out that Georg Northoff has indeed done a lot of work in linking phenomenology to the brain. In view of the reviewers comments we have reread the paper proposing the spatial temporal theory of consciousness in detail, and have cited this work in several places given its strong relevance to the present work. Of particular interest is that Northoff and colleagues propose that a specific temporo-spatial organisation is a fundamental distinguishing feature of different levels of consciousness. Our approach indeed does show that specific temporal properties do correlate with different levels of awareness. Please see our additions below in the discussion that highlights the specific theoretical links between such network dynamics and subjective experience.

Lines: 410-413

In accordance to this theory’s predictions, we show that consciousness is associated specific autocorrelation properties and a more complex dynamics repertoire which would theoretically support subjective feelings of continuity and fluctuations of states of consciousness over time (Northoff, Wainio-Theberge, & Evers, 2020).

In particular, we found strong convergence with the discussion in which we try to emphasise the additional complexity induced by the perspective of a state (within the intrinsic state space).

This is in fact brilliantly expressed in Northoff’s 2020 paper “Is temporo-spatial dynamics the “common currency” of brain and mind? In Quest of “Spatiotemporal Neuroscience”. We cite: “Crucially, on the current view, the very structure of this equation speaks to a *relational* perspective in space and time. This is because the rate of change of a state with time, in its state space,” p.38

This in fact supports the importance of **intrinsic dynamics**, described as a function of a state. This is very much in line with the theoretical foundations underlying this work, we have therefore inserted this in the introduction.

Lines: 95-97

In accordance with the spatiotemporal theory of consciousness, we focus on the intrinsic time and space constructed by the brain, enabling us to study the consciousness-relevant features of the resulting spatiotemporal landscape (Northoff & Huang, 2017; Northoff et al., 2020b).

As for the 2010 J cognitive neuroscience paper, we have only been able to find this: Methodological Pitfalls in the “Objective” Approach to Consciousness: Comments on Busch et al. (2009) by Morten Overgaard, Mads Jensen, and Kristian Sandberg

These authors in these comments, vocalise most of what we would like to answer in response to such claims. These are indeed very interesting problems, but explicit distinctions between what exactly is studied (e.g., in this case, phenomenal vs access consciousness), we feel, are of paramount importance.

- How about the global signal? A recent study by Tanabe et al. (2020) showed that the level of the global signal is parametrically related to the level/state of consciousness in various altered states. How do the authors deal with that? Ideally I would like to see their data with and without global signal regression....

This is indeed a very important study and speaks to the theory set out in the temporo-spatial theory of consciousness. We did not regress out the global signal because there is evidence it may change the relationships between brain regions (e.g., Dixon et al., 2017). Furthermore, as shown (Tanabe et al., 2020), the global signal is indeed a relevant signal to consciousness. However, given that we keep this effect constant across participants, and most importantly for the present paradigm, within participants, we felt this should not influence the construction of the intrinsic state space represented by the meta-matrix. Nonetheless, we appreciate the reviewer’s concerns and have processed the Cambridge anaesthesia and DOC dataset with and without global signal regression (GSR).

We found that, in comparison with the ACompCorr method, there was some reproducibility of results, however this was not great. In fact, out of all the reproductions with alternative methods, the use of global signal regression seemed to be the least successful in reproducing results. This does suggest that indeed the global signal is an important component of the signal which needs to be considered when evaluating consciousness, as suggested by Tanabe et al., 2020.

All of the following results are available through the supplementary materials.

When removing the global signal, in the whole brain PTSM analysis (**called TDSM in the latest version**), The linear and first exponential model show the same trend, but this was not the case in the second exponential model. Please see the below table with all the relevant statistics for the TDSM analysis.

(NETS= networks/granularity; SMLin=similarity of MM to linear model, SMexp= similarity to first exponential model (see methods); SMexp2= similarity of MM to the second exponential model; COEF= regression coefficient ; OR= odds ratio; CI_l= confidence interval (2.5%:97.5%) low, CI_h= confidence interval high; BRA_OMN= Brant’s omnia test (p<0.05= assumption violated))

Measure	NETS	OR	CI_l	CI_h	COEF	P	BRA_OMN
SMLin	126	0.645	0.392	1.041	-0.437	0.038	0.289
SMexp	126	0.702	0.432	1.126	-0.352	0.072	0.275
SMexp2	126	0.835	0.516	1.340	-0.179	0.228	0.333
SMLin	553	0.533	0.311	0.877	-0.628	0.007	0.398

SMexp	553	0.582	0.347	0.947	-0.541	0.016	0.434
SMexp2	553	0.697	0.424	1.132	-0.359	0.073	0.373

For the proximal transition complexity, we found that results did not reproduce in terms of the average rate of change (median and mean distance between each temporally sequential network state). However, the variation of the distance over time and the complexity (unpredictability, measured via ETC and sample entropy) showed a good degree of reproducibility. Please see the below table for the proximal analyses.

(STD= standard deviation; Shannon = Shannone entropy; SampEn = Sample entropy; ETC= effort to compress; COEF= regression coefficient ; OR= odds ratio; CI_l= confidence interval (2.5%:97.5%) low, CI_h= confidence interval high; BRA_OMN= Brant's omnia test (p<0.05= assumption violated); NETS= networks/granularity)

Measure	NETS	OR	CI_l	CI_h	COEF	P	BRA_OMN
Mean	126	1.405	2.303	1.142	-0.340	0.161	0.338
Median	126	1.392	2.302	1.151	-0.331	0.176	0.808
Mean	553	1.467	2.415	1.085	-0.383	0.113	0.497
Median	553	1.317	2.133	1.195	-0.275	0.242	0.453
STD	126	2.138	3.681	1.314	-0.759	0.003	0.427
Shanon	126	2.308	3.941	1.419	-0.836	0.001	0.621
STD	553	2.153	3.725	1.318	-0.767	0.003	0.314
Shanon	553	2.516	4.364	1.525	-0.922	0.0005	0.601
SampEn	126	2.100	1.309	3.509	0.742	0.002	0.847
ETC	126	1.987	1.229	3.407	0.686	0.007	0.476
SampEn	553	1.719	1.077	2.838	0.541	0.026	0.792
ETC	553	1.376	1.161	2.255	0.319	0.186	0.942

When investigating the complexity of the dMM with GSR, we found, overall, a lack of reproducibility (except when using the sample entropy with 13 removed timepoints, for the lower granularity parcellation). Note, the term distal parameter refers to the number of adjacent timepoints we remove to create the dMM (described in the main text). Please see below the table with results for the dMM analyses.

(SampEn = Sample entropy; ETC= effort to compress; COEF= regression coefficient; OR= odds ratio; CI_l= confidence interval (2.5%:97.5%) low, CI_h= confidence interval high; BRA_OMN= Brant's omnia test (p<0.05= assumption violated); NETS= networks/granularity)

Distal_parameter	MEAS	NETS	OR	CI_l	CI_h	COEF	P	BRA_OMN
13	SampEn	126	1.627	1.015	2.694	0.487	0.048	0.471
13	ETC	126	1.459	0.922	2.358	0.378	0.112	0.471
24	SampEn	126	1.139	0.714	1.833	0.131	0.584	0.153
24	ETC	126	1.215	0.785	1.920	0.195	0.389	0.210
13	SampEn	553	1.616	0.995	2.678	0.480	0.055	0.168
13	ETC	553	1.340	0.833	2.163	0.293	0.223	0.721
24	SampEn	553	1.199	0.747	1.938	0.182	0.451	0.015
24	ETC	553	1.369	0.878	2.178	0.314	0.172	0.006

This perhaps suggests that the global signal is particularly important for consciousness-related differences in the wider, (long-term/distal) state space of network states. However, of note is that

the inclusion of global signal had greater success in reproducing the dMM results when the cortex was considered on its own (available at https://github.com/Peter6789/Meta-Matrix_reproduction/blob/main/Subsystems_in_separate_OLRs/OUTPUT_SUBSYST_RAM_GS.csv), or as a covariate with the subcortex and the cortex (https://github.com/Peter6789/Meta-Matrix_reproduction/blob/main/Subsystems_in_same_OLR/OUTPUT_SUBSYST_SAME_RAM_GS.csv) . See “read me” file within the same GitHib folder as the CSV file showing subsystem specific results.

Although analyses with global signal regression reproduced some analyses, of note is that the effects were weaker. As Tanabe et al., suggest, it seems that removing the global signal may indeed be removing variance that is relevant to consciousness.

By removing the global signal, we may in fact be removing a signal which is important (but not fundamental, given reproducibility of some results) to understanding the specific individual’s intrinsic dynamics. Please see the differences in the MMs in supplementary material 2 for the same individual. Overall, the global signal regression results are presented and discussed in the supplementary materials (specifically S4, S6, S7, see S2 for a description)

- I would like to see the complexity applied to proximal series and the similarity to distal..but may be I overread that...

This comment had a substantial role in inspiring us to produce the new proximal analyses (see figure 2 below). For this we would like to express great appreciation to the reviewer. In fact, it was obvious to the authors since the start of this endeavour that in the MM there is both distal and proximal information that looks different (values close to the diagonal vs values far away from the diagonal). We attempted to investigate this in several ways. The reviewer’s comment motivated us to come up with a way to analyse proximal information.

We believe that these results provide more interpretable and specific characterisations of the network dynamics that are associated with consciousness (i.e., network dynamics are faster, less variable, but more unpredictable).

We tried several methods to characterise the proximal autocorrelational transitions between specific timepoints (minimum transition is 2s). We copied figure 2 in this document to facilitate reading.

A Proximal moment-to-moment changes in connectivity patterns

B Proximal transition timeseries on MM

C Average rate of change
OR=4.06 (p=0.0002)

D Standard Deviation of rate of change
OR=3.50 (p=0.00001)

E Sample entropy of rate of change
OR=4.41 (p=0.000006)

Please see below description from main text

Figure 2. Proximal network transitions; description and analyses.

(A) depicts the short-term transitions between each successive connectivity pattern (time =2s). The distance between these was measured via Pearson's correlation (see S2 for alternative distance metrics used). B shows where in the meta-matrix this information is represented (along the diagonal); shown by a green arrow. On the resulting timeseries, describing the rate of change of connectivity patterns over time, we calculated the average (C), the standard deviation (D) and the sample entropy (E; see S4 for alternative methods). These measures were inserted as an independent variable in ordinal logistic regression with conditions ordered according to presumed levels of awareness (control awake > sedation > minimally conscious state > unresponsive wakefulness syndrome). Con = control awake, SED=sedation; MCS=minimally conscious state; UWS= unresponsive wakefulness syndrome. OR= odds ratio. Red rhombi represent the mean, whilst blue triangles, the median.

Firstly, to investigate proximal transitions, we had to deal with the problem of how the data is obtained and organised, which has a bearing on the complexity measure used. If we take a timepoint's closest similarities we get effectively a timeseries (similarity of timepoint 1 to timepoint 2, 2 to 3, 3 to 4 etc.; shown in figure 2A and 2B). Therefore, we decided to apply measures to specific off-diagonals of the MM (green arrow in figure 2B). On this timeseries we applied several measures. Firstly, we calculated measures of the average (median reported in the main text, but reproduced with mean). The average similarity between successive timepoints indicates the extent to which a network changes over a specific time (e.g., 1TR, the specific temporal interval is discussed below). Another way to understand this is the "rate of change" of the network, or the speed of network dynamics. That is, lower average similarity between successive timepoints suggests that there was a tendency to travel greater "distances" during a specific time (we do not characterise the "direction" in which a network travels, although this may be a future endeavour and is mentioned at the end of the discussion). We found that, on average, networks dynamics were faster during higher levels of awareness. We then applied a measure of the variance of proximal transitions, namely standard deviation (we reproduced this with Shannon entropy). We also found, that, contrary to expectation, conscious conditions were associated with less variance than unconscious conditions. Furthermore, when we analysed the timepoint-to-timepoint temporal complexity (as measured by sample entropy and effort to compress), the results were the opposite than all other distribution measures (i.e., standard deviation, Shannon entropy). These complexity measures are different, in that they discount temporally sequential information and focus on the probability of different values appearing, independently of "where" in the timeseries they are found. The other complexity measures (effort-to-compress and sample entropy) instead, broadly, try to measure the degree of predictability of the next timepoint given the previous ones.

Please find below a new results subsection as found in the latest version of the paper.
Lines: 191-238

Proximal temporal complexity

We investigated to what extent shifts in sequential connectivity states that are temporally close to each other may be predictive of levels of awareness (Fig. 2A). This autocorrelational information is represented in the MM by the values close to the main diagonal (Fig. 2B, in green) which are characterised by high similarity. The distance (i.e., inverse of similarity; reproduced with other metrics) between each temporally successive connectivity pattern is represented (1 to 2; 2 to 3 etc.; see Fig. 1A) in the first subdiagonal (the second longest, just above/below the main diagonal). On this timeseries of short term connectivity pattern similarities, we calculated measures of central tendency (e.g., median), distribution breadth (e.g., standard deviation), and temporal complexity measures (i.e., sample entropy (Richman & Moorman, 2000); see S4 for alternative methods). We found that, on average, unconsciousness tended to be characterised by higher similarities in short temporal sequences (OR=4.06, C.I.1.7:9.97, $p=0.002$). This effect was particularly prominent in deep anaesthesia from the second dataset and in the UWS patients (Fig. 2C; see S4 for deep anaesthesia data). Thus, intrinsically defined network dynamics had a lower rate of change or speed (higher similarity=less distance/difference over time [2s]) in unconsciousness.

We then investigated the distribution of short-term temporal distances (Fig. 2D). Intriguingly, we found that the standard deviation of proximal time similarity values decreased with levels of awareness (OR=3.50, C.I.=2.02:6.60 $p=0.00001$), in contrast to what would be expected (Carhart-Harris et al., 2014). We interpreted this to signify that consciousness is characterised by smoother, more constant, intrinsic dynamic rates of change (~speed). To confirm this, we took the average of the absolute derivative of the proximal similarity values, indicating how the speed itself varied over time (i.e., whether rate of change accelerated/decelerated) and

found that it was highly correlated to the standard deviation of proximal time similarity values ($Rho=.84$; S4 for reproducibility). Thus, although on average faster, changes in network configurations tend to be, overall, more constant in consciousness (i.e., tend to not accelerate much).

Finally, we found that the temporal complexity (measured via sample entropy [(Richman & Moorman, 2000)]) of proximal timepoint network transitions increased with levels of awareness (OR:4.41, $p=0.000006$ C.I. 2.37:9.1; Fig. 2E). This indicates that the moment-to-moment transitions of connectivity states are more unpredictable in consciousness.

These results were robust across various measures and control analyses (S4) and lead to interesting characterisations of the temporally proximal transitions of networks during awareness. Whilst network transitions tend to be faster and have unpredictable temporal sequences, transitions overall tend to be smoother, with more constant rates of change. This suggests that consciousness is associated with network dynamics that are temporally complex, but nonetheless display a certain structure.

Of course, selecting the time between each proximal temporal transition (e.g., 1TR or 3TRs) in this investigation is arbitrary given the novelty of the method. Thus, we selected three different temporal intervals between proximal timepoint to be investigated: 1 (thus looking at only the immediate future, corresponding to 1 TR;), 6 (take the timeseries of similarities between timepoints that are 6TRs distance) and 13 (matching the dMM parameter). On these we calculate the relevant metrics on all off diagonals and average them. We find unequivocal reproducibility across this parameter in both alternative datasets. Results for this reproduction are discussed in supplementary material 4 and are available in full at https://github.com/Peter6789/Meta-Matrix_reproduction/tree/main/Proximal_analysis. Similarly, these results reproduced well across all methodological contingencies and datasets. There were exceptions for the central tendency measures when global signal regression or instantaneous phase synchrony was used. With this latter method, the complexity of temporal sequences measures (sample entropy and ETC) was only partially reproduced. All these reproductions are available through S4.

Nonetheless, the contradiction between the two complexity measures (e.g., Shannon entropy vs sample entropy) is very interesting, and would on the surface be contradictory to theoretical predictions (specifically entropic brain hypothesis as well as temporo-spatial theory). However, a deeper reading of these theories implies that consciousness is not associated with pure disorder, but a complex organisation. This may partially explain these results (figure 2D vs 2E)]. Please see below such thoughts as expressed in the discussion.

Lines: 416-425

Of particular interest is that whilst increased complexity of (proximal; Fig 2E and distal; Fig 3D) temporal sequences is in line with the entropic brain hypothesis (stating that consciousness leads to an increase in entropy (Carhart-Harris et al., 2014)), the increased breadth of proximal transitions (Fig 2D, which reproduced with Shannon entropy, see S4) would superficially seem to contrast this theory (see also Northoff et al., 2020). However, a deeper reading of this theory reveals that consciousness is acknowledged to be associated with an organisational effect which may function to reduce “surprising” states of the organism at different levels (biological, psychological, see (Carhart-Harris & Friston, 2010, 2019; K. Friston, 2010)). Therefore, the increased moment-to-moment unpredictability (Fig. 2E) but reduced variations in network transition speeds (Fig. 2D) may represent functional organisational properties that emerge with consciousness.

After we produced these results, we found that this measure was already published by Battaglia and colleagues (Battaglia et al., 2020), similarly to Varley and colleagues (although they focus on “activity” states) (Varley, Denny, Sporns, & Patania, 2021). Battaglia and colleagues, however, work under a different framework and investigate age-related effects rather than consciousness. This is duly acknowledged in the discussion.

Lines: 531-532

Furthermore, Battaglia and colleagues (Battaglia et al., 2020) developed a similar method of speed under a different framework to investigate how ageing relates to functional network dynamics.

Of note, this present approach may be united with clustering approaches used in the literature to characterise the “direction” of such dynamics. E.g., Acceleration towards a DMN state. This may prove a very interesting avenue of research. We discuss this briefly in the methodological considerations.

Lines: 543-545

Furthermore, uniting the present approach with clustering techniques (Demertzi et al., 2019; Huang, Zhang, Wu, Mashour, & Hudetz, 2020; Shine et al., 2019) may permit dynamical systems characterisations of specific (clustering-defined) states (e.g., acceleration towards a DMN state).

It should be noted that, it may not be entirely accurate to reduce such a high dimensional state space into distance from steady/fixed states (e.g., DMN, highly activated etc.). We tried to explicate the high dimensional nature of the distal dynamics (represented by distal dynamics dMM) in the main text and insert a new supplementary material to this end (S5).

The dMM (Fig. 3C) therefore defines how a temporally specific state is described by other states distal in time (Fig. 3A). It approximates the position of a specific connectivity state in a wider space of possible states, via the relationship (i.e., distance) to all other states. Whilst the proximal space dynamics are more easily understood (as time progresses similarity tends to decay; or distance tends to increase [see TSDM Fig. 1E]), the dMM may approximate a more complex and higher dimensional state space (e.g., see fig 1D) as it would describe the relative position of many different states in a complex space (see S5 for further characterisations). The difference between the proximal and distal space is corroborated by the difference in similarity values and their organisation (see differences in colour in different parts of the MM (e.g., Fig. 3B), also compare the TSDM (Fig. 1E) to the dMM (Fig. 3B and 3C)).

Given that we applied different measures (e.g., average, standard deviation) to proximal transitions, it would be reasonable to ask whether such measures would show interesting effects if applied to the dMM. We investigated this and found no consistent effects, although more data may provide more power for such analyses.

Lines: 284-287

We found that there was no appreciable difference in the average and variation of the similarity values represented in the dMM (S6). This suggests that with consciousness, there is no effect on the tendency to return to similar states, or in the variations of distal definitions of a state.

In response to the second part of the reviewer’s comments (i.e., “similarity to distal”), this is not at all straight forward. There is no model that could indeed produce such detailed intrinsic dynamics. In fact, the meta-matrix has been used before to evaluate which whole brain computational model

performs best (Deco et al., 2018; Deco, Tononi, Boly, & Kringelbach, 2015; Hansen, Battaglia, Spiegler, Deco, & Jirsa, 2015). In this approach the meta-matrix is the gold standard to evaluate the performance of dynamic connectivity simulations/models. Hence, there is no adequate model to produce such intrinsic dynamics (except those which have been evaluated via empirical intrinsic dynamics, as used in this study), as the use of computational models may lead to a certain degree of circularity.

- Introduction: line 80 – this is the protention-retention approach by Husserl...this was addressed by others like Northoff and the other studies of Tagliazuchi, Northoff, etc mentioned

We would like to thank the reviewer for their very insightful comment. Only after the reviewer's comment did we find that Northoff does indeed engage with Husserl. For our part we have been in correspondence with a phenomenologist who specialises in Gurwitsch (Yoshimi & Vinson, 2015). They have been over the manuscript more than once and was supportive.

In view of these highly relevant considerations, we have added the following section
Lines: 78-82

We base our approach on the phenomenological observation that any individual experience can be characterised by the experiences anteceding it and the experiences it may lead to (James, 1890; Nietzsche, 1886; Northoff, Wainio-Theberge, & Evers, 2020a; Tononi, Boly, Massimini, & Koch, 2016; Yoshimi & Vinson, 2015). Just as any neural state must be understood within its own systemic and dynamic context, we postulate the intra-individual context of any experiential state is foundational to its subjective value (James, 1890; Northoff & Stanghellini, 2016; Yoshimi & Vinson, 2015).

In regards to the Northoff paper from 2016 (Northoff & Stanghellini, 2016), we have been particularly interested in the extent we can assume that most of typical human consciousness is anchored to basic structures that are invariant. Such invariance is likely to exist, but may relate to very basic characterisations of consciousness (Damasio, 1998; Parvizi & Damasio, 2001). On the other hand, intrinsic temporo-spatial properties may indeed constitute some fundamental, invariant structures. However, this is beyond the scope of the present work, and thus we did not include these notions in our manuscript.

- My feeling is that there is a phenomenological overinterpretation: they measure the neural continuity with their autocorrelation-like analysis of functional connectivity pattern....but does that imply showing that the subjects really experience or do not experience a flow or stream of consciousness? That remains an inference and not implied by the data...Hence, there is no real convergence of dynamic and phenomenological features...as they do not have any subjective data...so the main issue is a new very well done analysis of functional connectivity patterns...

We very much appreciate this point. Although this analysis was inspired by phenomenological as well as theoretical observations, we do not provide evidence that such network dynamics are linked to phenomenological dynamics. We have thus reworded the discussion sections to ensure that we only suggest this is a possibility. In fact, we have been directly investigating the correspondence between experiential and network dynamics (with different experimental conditions) with some degree of preliminary success.

Please see below the changes made to remove phenomenological overinterpretation.

We have removed the following section

“We provide evidence for the common intuition that increasing levels of awareness are characterised by a rich “stream of consciousness” or flow of mental states (measured indirectly via the functional network variation intrinsic to the subject), that re-organise rapidly, are specific (informative) across longer periods of time and have more freedom from their structural underpinnings.”

We have instead modified the following section.

From:

“Given the consistent relationship between network connectivity and cognitive states (e.g., 9–18) the increase in MM complexity (indicating higher differentiation of connectivity states in time) may be interpreted as indicating a phenomenological increase in mental states and content in time, which may correspond to a wide exploration of the vernacular “stream of consciousness”.”

To:

Lines: 400-402

Given the consistent relationship between network connectivity and cognitive states (e.g., Andrews-Hanna et al., 2014; Christoff et al., 2009; Cole et al., 2016, 2013; Dixon et al., 2017, 2018; Fox et al., 2005; Margulies et al., 2016; Shine et al., 2019; Smith et al., 2009) these specific temporal network properties may be related to phenomenological dynamics of mental states and content (Carhart-Harris et al., 2014; Northoff & Huang, 2017; Tononi et al., 2016).

And as a future direction in relation to this point.

Lines: 538-542

Although the present study tried to intrinsically represent the stream of consciousness, we could not characterise the content of the stream of consciousness as each connectivity pattern was defined intrinsically to the resting state data via its similarity to all other patterns. The use of online experience sampling (Christoff et al., 2009) or naturalistic paradigms (e.g., audio, music, or movie watching (Naci et al., 2018), may provide a specification of the content of an individual’s stream of consciousness.

- The temporo-spatial theory of consciousness (TTC) should be discussed here....it fits much better the results than the IIT....also the paper by Lamme and Northoff 2020 may be important to cite as it compares different theories of consciousness...

This is a very fair point. The three theories that have inspired this work: the entropic brain hypothesis, information integration theory and the temporo-spatial theory of consciousness; we removed this latter for the submitted version to try and reduce the reader’s burden; after reviewing this theory again, we now realise this was a mistake and we thank the reviewer for pointing this out. We do this both in the introduction and discussion.

Please see text below from the revised version of the manuscript.

Lines: 93-101

This approach is related to several theoretical concepts that are relevant to the neuroscientific study of consciousness. In accordance with the spatiotemporal theory of consciousness, we focus on the intrinsic time and space constructed by the brain, enabling us to study the consciousness-relevant features of the resulting spatiotemporal landscape (Northoff & Huang, 2017; Northoff & Lamme, 2020; Northoff et al., 2020). The present approach is also related to theoretical concepts of the cause-effect repertoire of information integration theory (defined as the repertoire of all possible past and future states given the present state (Tononi et al., 2016)) and entropy (by which consciousness is characterised by high degrees of

information specificity and unpredictability; entropic brain hypothesis (Carhart-Harris et al., 2014), to define a proxy measure of the stream of consciousness.

In the discussion

Lines: 409-423

The temporo-spatial theory of consciousness (Northoff et al., 2017) postulates that the intrinsic temporal-spatial organisation of the brain is fundamental to the emergence of consciousness. In accordance to this theory's predictions, we show that consciousness is associated with specific autocorrelation properties and a more complex dynamics repertoire which would theoretically support subjective feelings of continuity and fluctuations of states of consciousness over time (Northoff, Wainio-Theberge, & Evers, 2020). Similarly, the approach set forth in this paper may permit to approximate an empirical measure of the size "cause effect repertoire" in Information integration theory terms, (i.e., the distribution of possible past and future states given the present state; thus defining "information" in this theory (Tononi et al., 2016)). Of particular interest is that whilst increased complexity of (proximal and distal) temporal sequences is in line with the entropic brain hypothesis (stating that consciousness leads to an increase in entropy (Carhart-Harris et al., 2014), the increased breadth of proximal transitions (which reproduced with Shannon entropy, see S4) would superficially seem to contrast this theory (see also (Northoff et al., 2020b)). However, a deeper reading of this theory reveals that consciousness is acknowledged to be associated with an organisational effect which ultimately functions to reduce "surprising" states of the organism at different levels (biological, psychological, see (Carhart-Harris & Friston, 2010, 2019; K. Friston, 2010)).

We have also added references to Northoff's work in the section below which suggests how the emergence of subjective experience may engender additional dynamics, as this work is very much relevant to this point.

Lines: 456-458

Thus, with the emergence of complex awareness, additional "apparent" changes (K. J. Friston, 1997) in the dynamic landscape may arise due to the structured, relative trajectory or "*perspective*" of the current brain state within its own intrinsic space of past and future states ((K. Friston, 2017, 2018; Northoff & Huang, 2017; Northoff et al., 2020b; Rabinovich, Huerta, Varona, & Afraimovich, 2008).

Reviewer #2 (Remarks to the Author):

In this article, the authors demonstrate the link between brain dynamic functional connectivity with theories of consciousness, providing solid evidence that the evolution of functional connectivity patterns over time is intrinsically related to consciousness level. The study is presented in a very clear way, targeting a broad audience in neuroscience, linking recent developments in neuroimaging and psychology to address fundamental aspects of brain function.

The authors propose rigorously validated measures that demonstrate robustness across datasets, parcellation schemes and methodological parameters (i.e., window length). Further, the results are shown not to be exclusive to the cortex and to replicate in subcortical structures and the cerebellum.

The technical implementation demonstrates expertise, keeping only the necessary degree of complexity inherent to the metrics employed, while maintaining a sufficient degree of interpretability.

However, some issues should be addressed to improve the quality of the manuscript before considering publication in Communications Biology.

We would like to kindly thank the reviewer for their kind and perceptive comments. We will endeavour to answer them all in an exhaustive manner.

General comments:

1 - 'We provide evidence for the common intuition that increasing levels of awareness are characterised by a rich "stream of consciousness" or flow of mental states (measured indirectly via the functional network variation intrinsic to the subject), that re-organise rapidly, are specific (informative) across longer periods of time and have more freedom from their structural underpinnings.'

Although the results presented herein reveal a clear relationship between the metrics employed and the levels of consciousness, the relationship with the 'stream of consciousness' can only be speculated, since it is unclear if and how the different connectivity patterns actually reflect mental states, so the authors could maybe rephrase to temper the claims.

We agree with the reviewer that this is speculative. The theoretical basis of this approach does try to provide a link between neural and subjective dynamics, as stated in the introduction. However, we do not have subjective measurements for these data and have therefore removed the claim above from the paper.

In the methodological considerations section of the discussion, we do say how this approach may be used to link neural and subjective dynamics. We are working on such a study now, with good preliminary results.

Lines: 538-542

Although the present study tried to intrinsically represent the stream of consciousness, we could not characterise the content of the stream of consciousness as each connectivity pattern was defined intrinsically to the resting state data via its similarity to all other patterns. The use of online experience sampling (Christoff et al., 2009) or naturalistic paradigms (e.g., audio, music, or movie watching (Naci et al., 2018)), may provide a specification of the content of an individual's stream of consciousness.

Most importantly, the authors should revise what is understood by 'rapid' (used several times throughout the text), since the results presented reflect only a stream occurring on a ultra-slow timescale (i.e. first 0.008 to 0.09 Hz band-pass filter + sliding 24s-window correlations). Indeed, it would really reinforce the validity of this work, and perhaps even improve the sensitivity of the measures, if the authors show that the results survive on faster evolutions of fMRI FC, which can still be detected in the fMRI signal.

We thank the reviewer for this excellent point. The specification of what we mean by rapid is paramount. However, we have significantly changed how we used rapid and speed in this paper, due to the addition of a new results sections which specifically investigates the average rate of change (or "speed", intended as distance over time) in network dynamics. The parameters for this (bandpass, sliding window length, TR) are constant across conditions. Hence there is a relative increase in speed of network dynamics in conscious conditions.

Nonetheless, we specify that the time on which we calculate distances in the new proximal temporal complexity section is 2 seconds.

Lines: 203-205

Thus, intrinsically defined network dynamics had a lower rate of change or speed (higher similarity=less distance/difference over time [2s]) in unconsciousness.

However, we also reproduce these results by averaging the metrics across several proximal transition times (i.e., 1 TR, 1 to 6 TRs and 1 to 13 TRs). We find results reproduce univocally and are available on github (https://github.com/Peter6789/Meta-Matrix_reproduction/tree/main/Proximal_analysis; relevant csv files end with *_OTHER_TIMEPOINTS) through supplementary material 4 (i.e., S4). Please see below the relevant section in regards to this, copied from the supplementary materials. See also below, for the use of a high pass filter in result reproducibility.

Reproducibility varying the proximal transition time (i.e., 2s, 12s and 26s)

Results reproduced when we looked at different temporal distances (the number of timepoint between two successive connectivity states; described in methods). This was repeated with the two anaesthesia datasets, respectively ordered with the DOC patients, using aCompCorr and sliding window. Replication occurred across all analyses. These are fully accessible in https://github.com/Peter6789/Meta-Matrix_reproduction/tree/main/Proximal_analysis.

The relevant CSV files end with *_OTHER_TIMEPOINTS.

Following the reviewer's comments, and the addition of the new results section we have removed all mentions of "rapid" in the results and discussion sections, and inserted speed in relation to this specific analysis. However, we do mention the rapidity of decay in the temporal decay of similarity models (TDSM; PTSM in previous version of the paper) in the methods. These have functions associated with this model, and therefore any interested reader could investigate the "rapidity of decay" if they so wished. Please see below the relevant section.

Lines:758-765

Given the novelty of this method, we created three TDSMs; one with linearly decreasing similarity over time, one with slow exponential decreasing similarity over time, and one with rapid exponential decreasing similarity over time. The reason for the exponential models is that they fit the data better (mean r value = 0.70 SD= 0.04) than the linear model (mean r value = 0.53 SD= 0.05). The slowly decreasing exponential model was created with the following matlab code: `exp(linspace(log(0.0001),log(1),n)/3)`, whilst the rapidly decreasing TDSM was created with `exp(linspace(log(0.0001),log(1),n)/1.5)`, n being the number of columns in the MM. The results were comparable between the linear and exponential TDSMs (S3).

In more detail, several works have shown that using the instantaneous phase of the BOLD signal allows the calculation of FC at the instantaneous level (Glerean et al, 2012), which significantly improves the results in comparative studies of time-versus-time dynamic FC matrices (i.e., Cabral et al., 2017). Basically, this allows calculating the Meta-Matrix without the smoothing induced by the sliding window, in order to verify how fast the 'stream of consciousness' can be detected. If this is not possible, at least the authors should mention this direction for future studies.

The reviewer is right in commenting that the sliding window analysis may also insert another confound, even though we used suggested parameters (Preti, Bolton, & Van De Ville, 2017). In accordance with the reviewer's suggestions, we tried reproducing all results using an instantaneous

phase synchrony method to obtain time varying connectivity matrices. Of note, is that according to the Bedrosiani requirement (Glerean, Salmi, Lahnakoski, Jääskeläinen, & Sams, 2012; Pedersen, Omidvarnia, Zalesky, & Jackson, 2018a), we used a very narrow bandpass filter for this analysis (0.03 to 0.07). We understand that this is to ensure that meaningful phases can be estimated from the Herbert transform.

In the methods section we write:

Lines:738-742

We also applied an alternative dynamic functional connectivity method (Glerean et al., 2012; Pedersen, Omidvarnia, Zalesky, & Jackson, 2018b), namely instantaneous phase synchrony (S2). This used a narrow band-pass filter (0.03:0.07 Hz) which is thought to satisfy Bedrosian’s requirement for phase synchrony analysis (Glerean et al., 2012; Pedersen et al., 2018b) (the smaller the bandwidth, the more likely meaningful phases can be estimated; however see also (Demertzi et al., 2019)).

For the most part, there seemed to be a convergence of results (with specific parameters).

For the reviewers ease we insert and discuss the whole-brain reproductions using instantaneous phase synchrony below. These are all accessible via the relevant supplementary materials which lead to specific folders within the github project (available at https://github.com/Peter6789/Meta-Matrix_reproduction).

For the TDSM analysis (PTSM in the previous version), there was a good degree of reproduction; however, the proportional odds assumption of the ordinal logistic regression was for the most part violated (Brant’s test $p < 0.05$). Please see below the table for the TDSM analysis.

(NETS= networks/granularity; SMLin=similarity of MM to linear model, SMexp= similarity to first exponential model (see methods); SMexp2= similarity of MM to the second exponential model; COEF= regression coefficient ; OR= odds ratio; CI_l= confidence interval (2.5%:97.5%) low, CI_h= confidence interval high; BRA_OMN= Brant’s omnia test ($p < 0.05$ = assumption violated))

MEAS	NETS	OR	CI_l	CI_h	COEF	P	BRA_OMN
SMLin	126	3.082	1.637	6.391	1.125	0.0006	0.000002
Smexp	126	2.808	1.537	5.630	1.033	0.0010	0.004
Smexp-2	126	2.039	1.230	3.550	0.712	0.0038	0.086
Smlin	553	3.549	1.783	8.283	1.267	0.0006	1.000
Smexp	553	3.314	1.781	6.848	1.198	0.0002	0.000001
Smexp-2	553	2.388	1.441	4.138	0.870	0.0005	0.011

For the proximal analyses results only reproduced for the variation and moment-to-moment temporal complexity, but not for the average rate of change. This may be explained by the use of a narrow band pass filter, or the effect that the instantaneous phase synchrony method may have at the edges of the timeseries (where one cannot properly estimate the phase). Of note is that the two ways to calculate this that we have investigated (Cabral, Kringelbach, & Deco, 2017; Pedersen et al., 2018a) both have such an effect at the edge of the timeseries, however this looks different when comparing the two methods. Although this is an interesting methodological issue, its detailed exploration is beyond the scope of this work. Please see below the table for the proximal analyses results.

(STD= standard deviation; Shannon = Shannone entropy; SampEn = Sample entropy; ETC= effort to compress; COEF= regression coefficient; OR= odds ratio; CI_l= confidence interval (2.5%:97.5%) low, CI_h= confidence interval high; BRA_OMN= Brant's omnia test (p<0.05= assumption violated); NETS= networks/granularity)

Measure	NETS	OR	CI_l	CI_h	COEF	P	BRA_OMN
Mean	126	1.320	1.255	2.201	0.277	0.281	0.792
Median	126	1.370	2.202	1.154	-0.314	0.183	0.816
Mean	553	1.548	1.094	2.950	0.437	0.122	0.280
Median	553	1.314	2.415	1.329	-0.273	0.359	0.883
STD	126	1.753	2.945	1.072	-0.562	0.028	0.903
Shanon	126	2.824	5.229	1.641	-1.038	0.0004	0.205
STD	553	1.867	3.204	1.124	-0.624	0.018	0.465
Shanon	553	3.866	8.054	2.080	-1.352	0.0001	0.414
SampEn	126	1.320	1.255	2.201	0.277	0.281	0.792
ETC	126	1.370	2.202	1.154	-0.314	0.183	0.816
SampEn	553	1.548	1.094	2.950	0.437	0.122	0.280
ETC	553	1.314	2.415	1.329	-0.273	0.359	0.883

Finally, the dMM results. These again, showed some degree of replication, but not excellent. Specifically, results failed to reproduce when we defined the dMM by cutting away the 24 closest points. However, results were convergent when we defined the dMM by removing the 13 closest points. Please see below showing the results for the dMM analyses.

(SampEn = Sample entropy; ETC= effort to compress; COEF= regression coefficient ; OR= odds ratio; CI_l= confidence interval (2.5%:97.5%) low, CI_h= confidence interval high; BRA_OMN= Brant's omnia test (p<0.05= assumption violated); NETS= networks/granularity)

Distal_parameter	MEAS	NETS	OR	CI_l	CI_h	COEF	P	BRA_OMN
13	SampEn	126	1.969	1.184	3.481	0.678	0.012	0.443
13	ETC	126	1.862	1.109	3.393	0.622	0.026	0.082
24	SampEn	126	1.410	0.877	2.285	0.344	0.155	0.635
24	ETC	126	1.410	0.852	2.361	0.343	0.175	0.598
13	SampEn	553	1.541	0.943	2.561	0.432	0.086	0.965
13	ETC	553	1.961	1.188	3.362	0.673	0.010	0.787
24	SampEn	553	1.227	0.508	1.311	-0.205	0.393	0.209
24	ETC	553	1.074	0.576	1.509	-0.072	0.769	0.649

For brevity, we will not insert the subsystem specific results (cortex, subcortex and cerebellum) here. But, when metrics for the subsystems were inserted individually in the OLR, there was a good level of significance all round (cerebellum just shy of significance similarly to the main text; https://github.com/Peter6789/Meta-Matrix_reproduction/blob/main/Subsystems_in_separate_OLRs/OUTPUT_SUBSYST_RAM_IPS.csv). When we inserted all the systems within the same OLR (https://github.com/Peter6789/Meta-Matrix_reproduction/blob/main/Subsystems_in_same_OLR/OUTPUT_SUBSYST_SAME_RAM_IPS.csv) as covariates, the cortex was close to reproducing results for the dMM ETC (p=0.06).

So overall, there seems to be some reproducibility of the results with the IPS method. However, this was not at all perfect. Future directions for this may be to use alternative methods that can resolve connectivity matrices at a TR resolution (Esfahlani et al., 2020), the additional manipulation of the

bandpass filter in IPS and other techniques (see Pedersen et al., 2018), the use of more data and the removal of timepoints at the edge of the timeseries. Of course, it would be interesting to apply our approach to EEG data. This would provide a much greater temporal resolution, and potentially more accurate estimations of the “speed” of state transitions (although note, we compare different conditions, thus looking at “relative” consciousness-specific changes in speed, rather than absolute speed). This may also provide an easy and cost-effective evaluative metrics for the clinic. We wrote this phrase in the methodological considerations.

Lines: 532-535

It is possible, with greater amounts of temporal data (e.g., electroencephalogram) that methods such as these may permit the study of the structure of an individual’s stream of consciousness in case studies and across specific experiential states.

2 - ‘We applied linear detrending and a 0.008 to 0.09 Hz band pass filter to eliminate scanner-related noise.’

This filter eliminates more than just scanner related noise, and removes also other physiological components, so the sentence should be corrected. If possible and feasible, it would be valuable if the authors could explore whether the results presented herein survive (or even improve), when considering a broader frequency range? For instance, recent studies have demonstrated that including physiological signal may have important contributions to resting-state activity, and therefore it would be important to evaluate if extending the frequency range improves the biomarking potential of the measures (Chen et al., 2020; Vohryzek et al., 2020). If this is not possible, at least the authors should mention it in the discussion.

We thank the reviewer for their comment. We agree that different frequency ranges may contain biologically relevant information. Therefore, as suggested by the reviewer, we repeated all the analyses by applying using a broader frequency range in the CAM-DOC analyses. In particular we applied a high pass filter (0.08:inf; similar to E.g., Spinder et al., 2021; and the minimally pre-processed human connectome project data).

More specifically, for the TDSM analysis (i.e., PTSM in previous version) we found that only the first exponential proximal time similarity model in the higher granularity whole brain model retained significance (please see below for the TDSM analyses). Please see below for the table illustrating TDSM analyses.

(NETS= networks/granularity; SMLin=similarity of MM to linear model, SMexp= similarity to first exponential model (see methods); SMexp2= similarity of MM to the second exponential model; COEF= regression coefficient; OR= odds ratio; CI_l= confidence interval (2.5%:97.5%) low, CI_h= confidence interval high; BRA_OMN= Brant’s omnia test (p<0.05= assumption violated))

MEAS	NETS	OR	CI_l	CI_h	COEF	P	BRA_OMN
SMLin	126	1.270	0.772	2.118	0.239	0.175	0.353
SMexp	126	1.335	0.803	2.244	0.289	0.134	0.552
SMexp-2	126	1.136	0.681	1.903	0.127	0.313	0.919
SMLin	553	1.504	0.897	2.567	0.408	0.063	0.181
SMexp	553	1.653	0.980	2.851	0.503	0.031	0.285
SMexp-2	553	1.344	0.800	2.284	0.295	0.133	0.666

Conversely, for the proximal analysis, there was a good degree of reproducibility. However, most central tendency measures had proportional odds assumptions violated (as shown by the Brant’s test). Please see below for the table with the proximal analyses.

(STD= standard deviation; Shannon = Shannone entropy; SampEn = Sample entropy; ETC= effort to compress; COEF= regression coefficient; OR= odds ratio; CI_l= confidence interval (2.5%:97.5%) low, CI_h= confidence interval high; BRA_OMN= Brant's omnia test (p<0.05= assumption violated); NETS= networks/granularity)

Measure	NETS	OR	CI_l	CI_h	COEF	P	BRA_OMN
Mean	126	1.949	1.117	3.922	0.668	0.033246	2.11E-08
Median	126	5.966	16.776	2.566	-1.786	0.000216	0.036
Mean	553	1.792	1.034	3.567	0.583	0.059349	0.004
Median	553	6.222	19.104	2.532	-1.828	0.000442	0.266
STD	126	62.736	580.872	9.453	-4.139	7.01E-05	0.173
Shanon	126	3.504	6.391	2.041	-1.254	1.45E-05	0.568
STD	553	44.470	333.021	8.100	-3.795	5.44E-05	0.177
Shanon	553	3.835	7.044	2.219	-1.344	4.39E-06	0.582
SampEn	126	4.504	2.454	9.083	1.505	5.70E-06	0.998
ETC	126	3.538	1.996	6.840	1.264	4.98E-05	0.739
SampEn	553	3.566	2.044	6.685	1.271	2.24E-05	0.950
ETC	553	3.567	2.002	6.885	1.272	4.74E-05	0.968

Noticeable is that the standard deviation of proximal transitions when the high pass filter is used yields incredibly high effects sizes (Odds ratio= 62 for lower granularity parcellation and 44 for higher granularity parcellation). Upon investigation we found that the high pass filtering results in outliers (Max z-score=5.5). We repeated the analysis removing all outliers and found results reproduced with sensible significant odds ratios (~3). It is difficult to say whether such high values are due to the pathology, consciousness-relevant dynamics (e.g., particularly impaired) or may be due to noise. Such an exploration is beyond the scope of the present work. Nonetheless, the fact that this alternative method reproduces results, with or without outliers, increases our confidence that this measure is sufficiently robust and is related to consciousness level.

For the whole brain complexity of the distal Meta-Matrix, there was consistent reproducibility of the results. However, some regressions violated proportional odds assumptions. Please see below for the table containing results for the dMM complexity in the whole brain.

(SampEn = Sample entropy; ETC= effort to compress; COEF= regression coefficient ; OR= odds ratio; CI_l= confidence interval (2.5%:97.5%) low, CI_h= confidence interval high; BRA_OMN= Brant's omnia test (p<0.05= assumption violated); NETS= networks/granularity)

Distal parameter	MEAS	NETS	OR	CI_l	CI_h	COEF	P	BRA_OMN
13	SampEn	126	3.087	1.782	5.737	1.127	0.0001	0.132
13	ETC	126	3.698	1.908	8.206	1.308	0.0004	0.327
24	SampEn	126	2.501	1.475	4.464	0.917	0.001	0.082
24	ETC	126	2.226	1.332	3.937	0.800	0.003	0.009
13	SampEn	553	2.929	1.712	5.314	1.075	0.0001	0.049
13	ETC	553	3.614	1.925	7.655	1.285	0.0002	0.766

24	SampEn	553	2.098	1.281	3.544	0.741	0.004	0.129
24	ETC	553	1.967	1.201	3.326	0.676	0.008	0.032

In the subsystems results (when each are inserted into the regression individually) we found a good degree of reproducibility. Similarly, to the bandpassed data, the cortex and subcortex were highly significant, whilst the cerebellum displayed trends. (results available at [https://github.com/Peter6789/Meta-](https://github.com/Peter6789/Meta-Matrix_reproduction/blob/main/Subsystems_in_separate_OLRs/OUTPUT_SUBSYST_RAM_HP.csv)

[Matrix_reproduction/blob/main/Subsystems_in_separate_OLRs/OUTPUT_SUBSYST_RAM_HP.csv](https://github.com/Peter6789/Meta-Matrix_reproduction/blob/main/Subsystems_in_separate_OLRs/OUTPUT_SUBSYST_RAM_HP.csv))

When we inserted the various subsystems into the same ordinal logistic regressions, we found the dMM complexity of the cortex to be independently predictive, similarly to the results reported in the main text.

3 - 'We calculated complexity measures (ETC and sample entropy) on each column of the distal meta-matrix and averaged across these.'

Why didn't the authors compute the complexity measures directly in the vectorized upper triangular part of the dMM matrix?

We would like to thank the reviewer for their insightful comment. We did realise that this in fact is an option. We thought that the implementation of the analysis in the manuscript is of more easy interpretation, in part given that sample entropy and ETC are sensitive to dynamic sequential information. Furthermore, the analysis of specific columns may permit a rather intuitive interpretation as to how complex that timepoint is in relation to its own past and future. We would like to thank the reviewer, as their comment did make us define the dMM with greater precision. We have added a figure (figure 3 in main text) for added clarity.

A Connectivity pattern's relationship to all other states distal in time

This shows how we understand the complexity of the wider dynamic space via "where" a timepoint specific state sits in relation to all other timepoints (we have also tried to give an intuition of the high dimensionality of this space in supplementary material 5 using a practical example).

Nonetheless, we agree with the reviewer that analysing the complexity of the whole vectorised matrix is an option. We ran this additional analysis and found that the dMM ETC and sample entropy results replicated in both datasets (available at https://github.com/Peter6789/Meta-Matrix_reproduction/tree/main/dMM_analysis; see *_VECTORISED_*.csv), although with tendentially smaller effect sizes.

Small corrections:

Figures:

Figure 1

Panel C – add colorbar;

panel D – add axis labels;

panel F Increase font sizes.

Legend for panel F is missing.

We would like to thank the reviewer for their suggestions. Please see below the most recent version of the figure (figure 1 in main text).

Figure 3 – A,B,C is in the legend but not in the figure.

Given the addition of new results, we have added a figure about the proximal analysis and removed “figure 3” that was in the previous version of the paper. We have reported information (including the novel analyses) about the subsystems in the text and through a table to make the manuscript more parsimonious.

Figure 4 – Missing title.

Also, it would be important to explain that the structural connectivity is obtained from applying tractography algorithms to DTI images.

Thank you for pointing this out. We have added as the title:

Complexity of structure-function dynamic relationship.

We have also specified we use tractography to construct structural connectivity in the figure legend.

Lines: 381-383

Illustration of method (A) for the sample entropy of the relationship between functional dynamic and tractography of diffusion tensor imaging data.

We have also inserted this information in the main text.

Lines: 330-332

We sought to further investigate brain dynamics in consciousness via the intrinsic relationship between dynamic functional connectivity and static structural connectivity (measured via tractography of diffusion tensor imaging).

Please revise the text in the following lines:

128-129 revise sentence

Thank you for the comment. Please see below the revised sentence.

Lines: 134-135

Subsequently, we calculated the similarity between each of the temporally-specific connectivity matrices (Fig. 1B).

155 widow

160 represent

304 suggest

305 ; and

363 references should be cited as numbers

Thank you kindly for this. We have corrected the spelling mistakes

Additional references:

Glerean, E., Salmi, J., Lahnakoski, J. M., Jääskeläinen, I. P., & Sams, M. (2012). Functional magnetic resonance imaging phase synchronization as a measure of dynamic functional connectivity. *Brain connectivity*, 2(2), 91-101.

Cabral, J., Vidaurre, D., Marques, P., Magalhães, R., Silva Moreira, P., Miguel Soares, J., Deco, G., Sousa, N., Kringelbach, M.L., 2017. Cognitive performance in healthy older adults relates to spontaneous switching between states of functional connectivity during rest. *Sci Rep* 7, 5135. <https://doi.org/10.1038/s41598-017-05425-7>

Chen, J.E., Lewis, L.D., Chang, C., Tian, Q., Fultz, N.E., Ohringer, N.A., Rosen, B.R., Polimeni, J.R., 2020. Resting-state “physiological networks.” *NeuroImage* 213,

116707. <https://doi.org/10.1016/j.neuroimage.2020.116707>

Vohryzek, J., Deco, G., Cessac, B., Kringelbach, M.L., Cabral, J., 2020. Ghost Attractors in Spontaneous Brain Activity: Recurrent Excursions Into Functionally-Relevant BOLD Phase-Locking States. *Front. Syst. Neurosci.* 14, 20. <https://doi.org/10.3389/fnsys.2020.00020>

Reviewer #3 (Remarks to the Author):

This paper proposed a method to investigate the temporal complexity of functional connectivity patterns, and showed that it scales with the level of awareness, and also demonstrated that the cortex, cerebellum and subcortex all display consciousness-relevant dynamics. Overall, I think this is a well-conducted study, which contributes to the field by providing a new method and an alternative perspective to study dynamic functional connectivity. However, I think the manuscript could be improved in the following aspects.

1. the similarity between connectivity matrices was assessed by Pearson's correlation. What is the rationale for this choice instead of other measures (e.g., a distance measure)? As the similarity is the basis to quantify complexity in this study, I'd suggest testing the reproducibility using alternative similarity methods.

We would like to thank the reviewer for their discerning comment. We agree that the distance metric is foundational to this approach and therefore deserves more attention. This approach, despite there being analogues in the literature (Cabral, Vidaurre, et al., 2017), was primarily inspired by representational similarity analysis (Kriegeskorte, 2008) which finds the distance between activity patterns via Pearson's correlation distance (Nili et al., 2014). However, this metric reduces two connectivity matrices to a number, and, the specific implementation of such a reduction may have a bearing on the results. Therefore, in consonance with the reviewer's suggestions, we reran all the analyses with the MM constructed via two alternative distance metrics. The first is Manhattan distance, also known as "City-Block distance". This metric is thought to deal well with high dimensionality data. Cabral et al., (Cabral, Vidaurre, et al., 2017) have also successfully constructed a MM using cosine distance and hence we also use this metric.

We found overall an excellent degree of reproducibility in the MM results. Here we will present the whole brain results for the reviewer's benefit.

For the TDSM analysis, The Manhattan distance reproduced results unambiguously.

Please see below the Manhattan Distance reproducibility of the TDSM results.

(NETS= networks/granularity; SMLin=similarity of MM to linear model, SMexp= similarity to first exponential model (see methods); SMexp2= similarity of MM to the second exponential model; COEF= regression coefficient ; OR= odds ratio; CI_l= confidence interval (2.5%:97.5%) low, CI_h= confidence interval high; BRA_OMN= Brant's omnia test (p<0.05= assumption violated))

MEAS	NETS	OR	CI_l	CI_h	COEF	P	BRA_OMN
SMLin	126	1.914	1.160	3.335	0.649	0.006	0.389
SMexp	126	2.065	1.234	3.727	0.725	0.004	0.163
SMexp-2	126	1.886	1.144	3.283	0.635	0.008	0.059
SMLin	553	1.898	1.144	3.357	0.641	0.008	0.469
SMexp	553	2.135	1.238	4.094	0.758	0.005	0.308
SMexp-2	553	1.941	1.131	3.697	0.663	0.012	0.016

Similarly, the Cosine distance reproduced all results.

Table with Cosine Distance reproducibility of the TDSM results.

MEAS	NETS	OR	CI_l	CI_h	COEF	P	BRA_OMN
------	------	----	------	------	------	---	---------

SMLin	126	1.898	1.144	3.357	0.641	0.008	0.469
SMexp	126	2.135	1.238	4.094	0.758	0.006	0.308
SMexp-2	126	1.941	1.131	3.697	0.663	0.013	0.016
SMLin	553	2.005	1.202	3.589	0.696	0.005	0.696
SMexp	553	2.281	1.313	4.389	0.824	0.003	0.470
SMexp-2	553	2.040	1.195	3.791	0.713	0.007	0.154

The proximal analyses were replicated perfectly.

Please see below results using Manhattan distance for the Proximal analyses

(STD= standard deviation; Shannon = Shannone entropy; SampEn = Sample entropy; ETC= effort to compress; COEF= regression coefficient ; OR= odds ratio; CI_l= confidence interval (2.5%:97.5%) low, CI_h= confidence interval high; BRA_OMN= Brant's omnia test (p<0.05= assumption violated); NETS= networks/granularity)

Measure	NETS	OR	CI_l	CI_h	COEF	P	BRA_OMN
Mean	126	4.273	9.667	2.199	-1.452	0.0001	0.301
Median	126	4.563	10.360	2.316	-1.518	0.00006	0.405
Mean	553	5.183	12.945	2.502	-1.645	0.00007	0.195
Median	553	5.020	12.050	2.458	-1.613	0.00006	0.261
STD	126	4.125	7.841	2.331	-1.417	0.000004	0.339
Shanon	126	3.475	6.322	2.024	-1.246	0.00001	0.546
STD	553	4.214	8.060	2.377	-1.438	0.000003	0.482
Shanon	553	3.396	6.156	1.987	-1.223	0.00002	0.878
SampEn	126	3.670	2.051	7.204	1.300	0.00004	0.876
ETC	126	2.037	1.250	3.482	0.711	0.005	0.877
SampEn	553	2.971	1.743	5.429	1.089	0.0001	0.977
ETC	553	2.009	1.237	3.415	0.698	0.006	0.512

Table with Cosine Distance for the Proximal analyses

Measure	NETS	OR	CI_l	CI_h	COEF	P	BRA_OMN
Mean	126	2.780	6.128	1.519	-1.022	0.003	2.38E-06
Median	126	3.459	7.572	1.844	-1.241	0.0005	0.456
Mean	553	4.457	13.510	1.972	-1.494	0.0025	0.254
Median	553	4.407	11.017	2.139	-1.483	0.0003	0.373
STD	126	3.276	6.037	1.913	-1.187	4.53E-05	0.059
Shanon	126	2.674	4.603	1.622	-0.984	0.0001	0.079
STD	553	3.483	6.538	2.013	-1.248	2.82E-05	0.137
Shanon	553	2.880	5.011	1.738	-1.058	8.06E-05	0.266
SampEn	126	4.165	2.246	8.537	1.427	2.51E-05	0.480
ETC	126	2.452	1.464	4.420	0.897	0.001	0.926
SampEn	553	3.187	1.820	6.025	1.159	0.0001	0.892
ETC	553	2.277	1.367	4.114	0.823	0.003	0.571

Table with Manhattan Distance for the dMM analyses

(SampEn = Sample entropy; ETC= effort to compress; COEF= regression coefficient ; OR= odds ratio; CI_l= confidence interval (2.5%:97.5%) low, CI_h= confidence interval high; BRA_OMN= Brant's omnia test (p<0.05= assumption violated); NETS= networks/granularity)

Distal_parameter	MEAS	NETS	OR	CI_l	CI_h	COEF	P	BRA_OMN
13	SampEn	126	2.228	1.337	3.950	0.801	0.0034	0.435
13	ETC	126	2.700	1.578	5.072	0.993	0.0007	0.880
24	SampEn	126	2.023	1.237	3.452	0.705	0.0066	0.279
24	ETC	126	2.329	1.419	4.112	0.845	0.0016	0.600
13	SampEn	553	2.923	1.682	5.537	1.072	0.0004	0.636
13	ETC	553	3.736	2.022	7.956	1.318	0.0001	0.306
24	SampEn	553	2.702	1.593	4.901	0.994	0.0005	0.391
24	ETC	553	3.211	1.823	6.376	1.167	0.0002	0.078

Table with Cosine Distance for the dMM analyses

Distal_parameter	MEAS	NETS	OR	CI_l	CI_h	COEF	P	BRA_OMN
13	SampEn	126	2.923	1.682	5.537	1.072	0.0003	0.636
13	ETC	126	3.736	2.022	7.956	1.318	0.0001	0.306
24	SampEn	126	2.702	1.593	4.901	0.994	0.0004	0.391
24	ETC	126	3.211	1.823	6.376	1.167	0.0002	0.078
13	SampEn	553	2.946	1.674	5.730	1.081	0.0005	0.481
13	ETC	553	3.655	1.989	7.641	1.296	0.0001	0.381
24	SampEn	553	2.732	1.590	5.096	1.005	0.0006	0.396
24	ETC	553	3.239	1.828	6.442	1.175	0.0002	0.103

These distance metrics also reproduce results when the cortex, subcortex and cerebellum are inserted independently in separate regressions and as covariates in the same regression.

To view subsystem specific results, please see

Cosine distance, when subsystems are inserted independently:

[https://github.com/Peter6789/Meta-](https://github.com/Peter6789/Meta-Matrix_reproduction/blob/main/Subsystems_in_separate_OLRs/OUTPUT_SUBSYST_RAM_COS.csv)

[Matrix_reproduction/blob/main/Subsystems_in_separate_OLRs/OUTPUT_SUBSYST_RAM_COS.csv](https://github.com/Peter6789/Meta-Matrix_reproduction/blob/main/Subsystems_in_separate_OLRs/OUTPUT_SUBSYST_RAM_COS.csv)

Manhattan distance, when subsystems are inserted independently:

[https://github.com/Peter6789/Meta-](https://github.com/Peter6789/Meta-Matrix_reproduction/blob/main/Subsystems_in_separate_OLRs/OUTPUT_SUBSYST_RAM_CB.csv)

[Matrix_reproduction/blob/main/Subsystems_in_separate_OLRs/OUTPUT_SUBSYST_RAM_CB.csv](https://github.com/Peter6789/Meta-Matrix_reproduction/blob/main/Subsystems_in_separate_OLRs/OUTPUT_SUBSYST_RAM_CB.csv)

Cosine distance, subsystems inserted as covariates: [https://github.com/Peter6789/Meta-](https://github.com/Peter6789/Meta-Matrix_reproduction/blob/main/Subsystems_in_same_OLR/OUTPUT_SUBSYST_SAME_RAM_COS.csv)

[Matrix_reproduction/blob/main/Subsystems_in_same_OLR/OUTPUT_SUBSYST_SAME_RAM_COS.csv](https://github.com/Peter6789/Meta-Matrix_reproduction/blob/main/Subsystems_in_same_OLR/OUTPUT_SUBSYST_SAME_RAM_COS.csv)

[v](https://github.com/Peter6789/Meta-Matrix_reproduction/blob/main/Subsystems_in_same_OLR/OUTPUT_SUBSYST_SAME_RAM_CB.csv)

Manhattan distance, subsystems inserted as covariates: [https://github.com/Peter6789/Meta-](https://github.com/Peter6789/Meta-Matrix_reproduction/blob/main/Subsystems_in_same_OLR/OUTPUT_SUBSYST_SAME_RAM_CB.csv)

[Matrix_reproduction/blob/main/Subsystems_in_same_OLR/OUTPUT_SUBSYST_SAME_RAM_CB.csv](https://github.com/Peter6789/Meta-Matrix_reproduction/blob/main/Subsystems_in_same_OLR/OUTPUT_SUBSYST_SAME_RAM_CB.csv)

Furthermore, we reproduced the subsystem specific results for the structural functional analysis using these two alternative distance metrics. We therefore compared them using Cosine and Manhattan distance metrics.

Interestingly we found that the Cosine metric reproduced all analyses, whilst the Manhattan metric only reproduced the Subcortical results. These results are presented in S11.

2. Could you provide and discuss the pairwise comparison (as did in structural-functional similarity section Line 296-303) results for Figure 1-3? In most of the pTSM cases and some dMM cases, it looks like no obvious difference between consciousness and sedation states, despite of significant odd ratios across all states.

We would like to thank the reviewer for their comment. We tried to avoid pairwise comparisons, in part due to the multiple comparisons problem we would encounter. Furthermore, if we were to run these results in all the attempted replications (and datasets) we would run into hundreds if not thousands of analyses which would lengthen the already substantial paper considerably. We in fact adopted the ordinal logistic regression analysis as this permitted us to test the hypothesis, “does the metric scale with presumed levels of awareness” without many assumptions (e.g., linearity, distance between conditions etc.). in the structural functional similarity section, we do not test the difference in the complexity of the dynamic relationship (shown in figure 4B), but the differences in maximum similarity between the structural and functional connectivity matrices. We performed this analysis as we believed that this would increase the interpretability of the results. This permitted us to say, that not only the structural functional relationship was more dynamically complex, but DOC patients tended to have moments where their functional states were more similar to the structural states, reproducing in a novel manner previous results (Barttfeld et al., 2015). Thus, we could speak of a relative structural-functional “untethering” in the discussion, which was particularly interesting for the meta-stability and emergence discussion.

Nonetheless, we appreciate the reviewers concern, and in fact, share their curiosity. Therefore, we have computed the Mann-Whitney U test for the results and inserted them below for the reviewer’s benefit. We decided to do planned comparisons according to our interest to reduce the number of multiple comparisons. That is the difference between the control condition and all other conditions, as well as the difference between the DOC conditions (given clinical relevance). If we correct multiple comparisons within each type of analysis the new alpha value is 0.0125

For the whole brain 126 parcellation with the first exponential linear model

CON>UWS W=30, p=0.0005
CON>MCS W=50, p=0.027
CON>SED W=143, p=0.56
MCS>UWS W= 25, p=0.010

For the proximal analyses, Central Tendency measure (Median), for lower granularity parcellation

CON>UWS W=24, p=0.0001
CON>MCS W=71, p=0.22
CON>SED W=112, p=0.11
MCS>UWS W= 33, p=0.043

For the standard deviation of proximal transitions

CON>UWS W=17, p=0.00002
CON>MCS W=19, p=0.0001
CON>SED W=54, p=0.0003

MCS>UWS W= 64, p=0.92

For the sample entropy of proximal transitions

CON>UWS W=188, p=0.0003

CON>MCS W=143, p=0.043

CON>SED W=215, p=0.09

MCS>UWS W= 91, p=0.13

For the distal analysis, ETC, distal parameter = 13. Lower granularity parcellation.

CON>UWS W=197, p=0.00005

CON>MCS W=140.5, p=0.06

CON>SED W=213, p=0.11

MCS>UWS W= 102, p=0.02

These are interesting results. However, we are concerned that the addition of such results would only place more burden on the reader, and perhaps dilute the “take home messages” of this study. Furthermore, due to the important implications that such differences may have (for example; what measures can distinguish MCS and UWS), the reproducibility of such results should be dutifully checked across all methodological contingencies (parcellation, datasets, bandpass, dynamic FC method, alternative metrics etc.). This, as discussed, would result in hundreds if not thousands of analyses, which we believe is beyond the scope of the present work (having already checked the results in the main text extensively).

3. Comparing the results of whole brain, cortex, subcortex, and cerebellum, I wonder whether odd ratios to some extent can imply the performance of complexity in prediction of conscious state? For example, in Figure 4, OR is higher for cortex than subcortex and cerebellum; but how to understand the OR is the lowest among the four?

Absolutely, the OR can be considered a measure of effect size. Therefore, it is possible to compare the different ORs. There is also another metric which permits to take into account the “parsimoniousness” of the ordinal logistic regression. However, given the many different ordinal logistic regressions that would need evaluating (especially if we include different parcellations, alternative measure and alternative methodologies), we believe this is not too feasible.

However, we do use the OR to decide which proximal and distal metrics to insert as covariates (partly inspired by the reviewers’ comment above and below), when we investigate whether these different metrics have independent predictive power. Due to a great degree of variation across reproducibility attempts, we cannot conclude anything firmly. Please see the relevant section below.

Lines: 296-309

Relationship between distal and proximal measures.

We found that with consciousness, proximal transitions tend to be less predictable, quicker, but more constant. Furthermore, the wider dynamic state space (approximated by the dMM), seems to be more complex (“less compressible”) in consciousness. We investigated to what extent these measures held unique variance. We found that, all variables were correlated to each other ($p < 0.001$), although there seemed to be evidence the different measures displayed some independent variance (Fig. 3E, averaged explained variance 22%), specifically between proximal (Fig 2C, 2D, and 2E) and distal measures (fig 3D; see S7 for reproducibility). We then

inserted the variables with the highest effect sizes (odds ratios) for the proximal and distal measures into the same ordinal logistic regression as covariates (namely sample entropy of proximal transitions and compressibility of dMM). We found that, there was evidence each of these had independent predictive power (dMM compressibility OR= 3.82, $p=0.0004$; Proximal SD OR=1.93, $p=0.02$), although this did not reproduce unequivocally across different control analyses (S7 for details).

Nonetheless, higher odds ratio suggests better predictive power, whilst lower ones less predictive power.

4. The results suggest that cortex, cerebellum and subcortex all display consciousness-relevant dynamics. I think it would be helpful if the authors could further explore (use a statistical method?) whether subcortex and cerebellum could provide additional information in the prediction of conscious state beyond cortex

This is an excellent point. We agree that it would be informative to find out whether the different subsystems (i.e., cortex, subcortex, cerebellum) have independent predictive power when using such metrics. We therefore inserted the metrics for the cortex, subcortex and cerebellum within the same ordinal logistic regression as covariates. We ran an OLR for each measure (central tendency, standard deviation and sample entropy for the proximal transitions, and ETC for the dMM and structure to function dynamic complexity).

We ran this for all methodological contingencies and found that only one effect was consistent, that is the dMM complexity of the cortex. Please see the section below, as it appears in the revised version of the manuscript.

Lines: 316-324

We then sought to investigate whether these cytoarchitecturally distinct systems had unique predictive power. We therefore inserted the intrinsic dynamic properties of the cortex, subcortex and cerebellum as covariates in the same ordinal logistic regression. We ran a separate ordinal logistic regression for each measure, namely the average, standard deviation and sample entropy for the proximal transitions and the compressibility of the dMM.

We found that the only effect that consistently reproduced across analyses (described in S9) was the dMM compressibility of the cortex, whilst the significance of the temporal properties of the subcortex and cerebellum seemed to depend of pre-processing contingencies. (S2 for description of preprocessing contingencies; and S9 for results).

Of note is that in some methodological contingencies the subcortex and the cerebellum were significant in certain measures. However, we do not go into this in depth given that these results are not stable.

For the dynamic complexity of the structure function relationship, we instead found that both the cortex and the subcortex had independent predictive power. Of note is that the Manhattan distance replicated this analysis only for the subcortex (similarly to when these variables are inserted into separate ordinal logistic regression), whilst the cosine distance displayed the same results. Please see below the relevant section added to the results.

Lines: 357-359

Interestingly, when the complexity of the structure-function dynamic relationship for each subsystem was inserted in the same OLR, both the cortex ($p=0.01$) and the subcortex

($p=0.0003$) seemed to retain independent predictive power, whilst the cerebellum was not significant ($p=0.07$; see S10 and S11 for various reproductions).

We decided to not include the Odds ratios in the main text because these vary widely between different reproducibility attempts. Hence, we did not want to mislead the reader, or add too many lines of explanation.

5. Figure 1. The matrices in C and E should be represented in the same size if they are. Please indicate which model was used in E

Thank you for this comment. We agree that such changes would increase the clarity of the manuscript. Hence, we have implemented them. Please see below the added text to the figure caption.

Lines: 181-184

We tested the predictability of the whole MM by comparing the meta-matrix to the temporal decay of similarity model (TDSM) (E), in which timepoints that are closer to each other are more similar (three of these models constructed, see methods; first exponential decay model presented in this figure).

6. Figure 2. please specify the measures and parameters used in the figure (12 or 24 timepoints, ETC or sample entropy), instead of describing the method in general

Again, this is a valid point. We created the figure in powerpoint for illustration purposes only and does not have a specific parameter. However, the results presented use the parameter of 13, which we have added to the figure caption.

7. Figure 4. Kolmogorov complexity measures?

We appreciate this may be confusing, so we have removed this. See “entropy measures” methods section for more information about this.

References

- Andrews-Hanna, J. R., Smallwood, J., & Spreng, R. N. (2014). The default network and self-generated thought: Component processes, dynamic control, and clinical relevance. *Annals of the New York Academy of Sciences*, *1316*(1), 29–52. <https://doi.org/10.1111/nyas.12360>
- Barttfeld, P., Uhrig, L., Sitt, J. D., Sigman, M., Jarraya, B., & Dehaene, S. (2015). Signature of consciousness in the dynamics of resting-state brain activity. *Proceedings of the National Academy of Sciences*, *112*(3), 887–892. <https://doi.org/10.1073/pnas.1418031112>
- Battaglia, D., Boudou, T., Hansen, E. C. A., Lombardo, D., Chettouf, S., Daffertshofer, A., ... Jirsa, V. (2020). Dynamic Functional Connectivity between order and randomness and its evolution across the human adult lifespan. *NeuroImage*, *222*(May), 117156. <https://doi.org/10.1016/j.neuroimage.2020.117156>
- Boly, M., Sasai, S., Gosseries, O., Oizumi, M., Casali, A., Massimini, M., & Tononi, G. (2015). Stimulus set meaningfulness and neurophysiological differentiation: A functional magnetic resonance imaging study. *PLoS ONE*, *10*(5). <https://doi.org/10.1371/journal.pone.0125337>
- Cabral, J., Kringelbach, M. L., & Deco, G. (2017). Functional connectivity dynamically evolves on multiple time-scales over a static structural connectome: Models and mechanisms.

- NeuroImage*, 160(March), 84–96. <https://doi.org/10.1016/j.neuroimage.2017.03.045>
- Cabral, J., Vidaurre, D., Marques, P., Magalhães, R., Silva Moreira, P., Miguel Soares, J., ... Kringelbach, M. L. (2017). Cognitive performance in healthy older adults relates to spontaneous switching between states of functional connectivity during rest. *Scientific Reports*, 7(1), 1–13. <https://doi.org/10.1038/s41598-017-05425-7>
- Carhart-Harris, R. L., & Friston, K. J. (2010). The default-mode, ego-functions and free-energy: A neurobiological account of Freudian ideas. *Brain*, 133(4), 1265–1283. <https://doi.org/10.1093/brain/awq010>
- Carhart-Harris, R. L., & Friston, K. J. (2019). REBUS and the anarchic brain: Toward a unified model of the brain action of psychedelics. *Pharmacological Reviews*, 71(3), 316–344. <https://doi.org/10.1124/pr.118.017160>
- Carhart-Harris, R. L., Leech, R., Hellyer, P. J., Shanahan, M., Feilding, A., Tagliazucchi, E., ... Nutt, D. (2014). The entropic brain: A theory of conscious states informed by neuroimaging research with psychedelic drugs. *Frontiers in Human Neuroscience*, 8(1 FEB), 1–22. <https://doi.org/10.3389/fnhum.2014.00020>
- Christoff, K., Gordon, A. M., Smallwood, J., Smith, R., & Schooler, J. W. (2009). Experience sampling during fMRI reveals default network and executive system contributions to mind wandering. *Proceedings of the National Academy of Sciences of the United States of America*, 106(21), 8719–8724. <https://doi.org/10.1073/pnas.0900234106>
- Cole, M. W., Ito, T., Bassett, D. S., & Schultz, D. H. (2016). Activity flow over resting-state networks shapes cognitive task activations. *Nature Neuroscience*, 19(12), 1718–1726. <https://doi.org/10.1038/nn.4406>
- Cole, M. W., Reynolds, J. R., Power, J. D., Repovs, G., Anticevic, A., & Braver, T. S. (2013). Multi-task connectivity reveals flexible hubs for adaptive task control. *Nature Neuroscience*, 16(9), 1348–1355. <https://doi.org/10.1038/nn.3470>
- Coppola, P., Spindler, L. R. B., Luppi, A. I., Adapa, R., Naci, L., Allanson, J., ... Stamatakis, E. A. (2022). Network dynamics scale with levels of awareness. *NeuroImage*, 254(March), 119128. <https://doi.org/10.1016/j.neuroimage.2022.119128>
- Damasio, A. R. (1998). Investigating the biology of consciousness. *Philosophical Transactions of the Royal Society B: Biological Sciences*, 353(1377), 1879–1882. <https://doi.org/10.1098/rstb.1998.0339>
- Deco, G., Cruzat, J., Cabral, J., Knudsen, G. M., Carhart-Harris, R. L., Whybrow, P. C., ... Kringelbach, M. L. (2018). Whole-Brain Multimodal Neuroimaging Model Using Serotonin Receptor Maps Explains Non-linear Functional Effects of LSD. *Current Biology*, 28(19), 3065–3074.e6. <https://doi.org/10.1016/j.cub.2018.07.083>
- Deco, G., Tononi, G., Boly, M., & Kringelbach, M. L. (2015). Rethinking segregation and integration : contributions of whole-brain modelling, (June 2016). <https://doi.org/10.1038/nrn3963>
- Delgado-Bonal, A., & Marshak, A. (2019). *Approximate entropy and sample entropy: A comprehensive tutorial*. *Entropy* (Vol. 21). <https://doi.org/10.3390/e21060541>
- Demertzi, A., Tagliazucchi, E., Dehaene, S., Deco, G., Barttfeld, P., Raimondo, F., ... Sitt, J. D. (2019). Human consciousness is supported by dynamic complex patterns of brain signal coordination. *Science Advances*, 5(2). <https://doi.org/10.1126/sciadv.aat7603>
- Dixon, M. L., Andrews-Hanna, J. R., Spreng, R. N., Irving, Z. C., Mills, C., Girn, M., & Christoff, K.

- (2017). Interactions between the default network and dorsal attention network vary across default subsystems, time, and cognitive states. *NeuroImage*, *147*(December 2016), 632–649. <https://doi.org/10.1016/j.neuroimage.2016.12.073>
- Dixon, M. L., De La Vega, A., Mills, C., Andrews-Hanna, J., Spreng, R. N., Cole, M. W., & Christoff, K. (2018). Heterogeneity within the frontoparietal control network and its relationship to the default and dorsal attention networks. *Proceedings of the National Academy of Sciences*, *115*(7), E1598–E1607. <https://doi.org/10.1073/pnas.1715766115>
- Esfahlani, F. Z., Jo, Y., Faskowitz, J., Byrge, L., Kennedy, D. P., Sporns, O., & Betzel, R. F. (2020). High-amplitude cofluctuations in cortical activity drive functional connectivity. *Proceedings of the National Academy of Sciences of the United States of America*, *117*(45), 28393–28401. <https://doi.org/10.1073/pnas.2005531117>
- Fox, M. D., Snyder, A. Z., Vincent, J. L., Corbetta, M., Van Essen, D. C., & Raichle, M. E. (2005). The human brain is intrinsically organized into dynamic, anticorrelated functional networks. *Proceedings of the National Academy of Sciences of the United States of America*, *102*(27), 9673–9678. <https://doi.org/10.1073/pnas.0504136102>
- Friston, K. (2010). The free-energy principle: a unified brain theory? *Nature Reviews. Neuroscience*, *11*(2), 127–138. <https://doi.org/10.1038/nrn2787>
- Friston, K. (2017). The Mathematics of Mind-Time. *Aeon*, 1–9.
- Friston, K. (2018). Am i self-conscious? (or does self-organization entail self-consciousness?). *Frontiers in Psychology*, *9*(APR), 1–10. <https://doi.org/10.3389/fpsyg.2018.00579>
- Friston, K. J. (1997). Transients, metastability, and neuronal dynamics. *NeuroImage*, *5*(2), 164–171. <https://doi.org/10.1006/nimg.1997.0259>
- Glerean, E., Salmi, J., Lahnakoski, J. M., Jääskeläinen, I. P., & Sams, M. (2012). Functional Magnetic Resonance Imaging Phase Synchronization as a Measure of Dynamic Functional Connectivity. *Brain Connectivity*, *2*(2), 91–101. <https://doi.org/10.1089/brain.2011.0068>
- Hansen, E. C. A., Battaglia, D., Spiegler, A., Deco, G., & Jirsa, V. K. (2015). Functional connectivity dynamics: Modeling the switching behavior of the resting state. *NeuroImage*, *105*, 525–535. <https://doi.org/10.1016/j.neuroimage.2014.11.001>
- Huang, Z., Tarnal, V., Vlisides, P. E., Janke, E. L., McKinney, A. M., Picton, P., ... Hudetz, A. G. (2021). Asymmetric neural dynamics characterize loss and recovery of consciousness. *NeuroImage*, *236*(March), 118042. <https://doi.org/10.1016/j.neuroimage.2021.118042>
- Huang, Z., Zhang, J., Wu, J., Mashour, G. A., & Hudetz, A. G. (2020). Temporal circuit of macroscale dynamic brain activity supports human consciousness. *Science Advances*, *6*(11), 1–15. <https://doi.org/10.1126/sciadv.aaz0087>
- James, W. (1890). *The Principles of Psychology*, 4004.
- Kolmogorov. (1965). Three approaches to the quantitative definition of information. *Problemy Peredachi Informatsi*, *1*(1), 3–11.
- Kriegeskorte, N. (2008). Representational similarity analysis – connecting the branches of systems neuroscience. *Frontiers in Systems Neuroscience*, *2*(November), 1–28. <https://doi.org/10.3389/neuro.06.004.2008>
- Lake, D. E., Richman, J. S., Pamela Griffin, M., & Randall Moorman, J. (2002). Sample entropy analysis of neonatal heart rate variability. *American Journal of Physiology - Regulatory Integrative and Comparative Physiology*, *283*(3 52-3), 789–797. <https://doi.org/10.1152/ajpregu.00069.2002>

- Liu, X., Ward, B. D., Binder, J. R., Li, S. J., & Hudetz, A. G. (2014). Scale-free functional connectivity of the brain is maintained in anesthetized healthy participants but not in patients with unresponsive wakefulness syndrome. *PLoS ONE*, *9*(3).
<https://doi.org/10.1371/journal.pone.0092182>
- Luppi, A. I., Craig, M. M., Pappas, I., Finoia, P., Williams, G. B., Allanson, J., ... Stamatakis, E. A. (2019). Consciousness-specific dynamic interactions of brain integration and functional diversity. *Nature Communications*, *10*(1). <https://doi.org/10.1038/s41467-019-12658-9>
- Margulies, D. S., Ghosh, S. S., Goulas, A., Falkiewicz, M., Huntenburg, J. M., Langs, G., ... Smallwood, J. (2016). Situating the default-mode network along a principal gradient of macroscale cortical organization. *Proceedings of the National Academy of Sciences*, *113*(44), 12574–12579.
<https://doi.org/10.1073/pnas.1608282113>
- Mitchell, M. (2011). *Complexity. A Guided Tour* (1st ed.). Oxford University Press.
- Naci, L., Haugg, A., MacDonald, A., Anello, M., Houldin, E., Naqshbandi, S., ... Owen, A. M. (2018). Functional diversity of brain networks supports consciousness and verbal intelligence. *Scientific Reports*, *8*(1), 1–15. <https://doi.org/10.1038/s41598-018-31525-z>
- Nagaraj, N., Balasubramanian, K., & Dey, S. (2013). A new complexity measure for time series analysis and classification. *European Physical Journal: Special Topics*, *222*(3–4), 847–860.
<https://doi.org/10.1140/epjst/e2013-01888-9>
- Nietzsche, F. (1886). *Beyond Good and Evil*. Cambridge University Press.
- Nili, H., Wingfield, C., Walther, A., Su, L., Marslen-Wilson, W., & Kriegeskorte, N. (2014). A Toolbox for Representational Similarity Analysis. *PLoS Computational Biology*, *10*(4).
<https://doi.org/10.1371/journal.pcbi.1003553>
- Northoff, G., & Huang, Z. (2017). How do the brain's time and space mediate consciousness and its different dimensions? Temporo-spatial theory of consciousness (TTC). *Neuroscience and Biobehavioral Reviews*, *80*(May), 630–645. <https://doi.org/10.1016/j.neubiorev.2017.07.013>
- Northoff, G., & Lamme, V. (2020). Neural signs and mechanisms of consciousness: Is there a potential convergence of theories of consciousness in sight? *Neuroscience and Biobehavioral Reviews*, *118*(May), 568–587. <https://doi.org/10.1016/j.neubiorev.2020.07.019>
- Northoff, G., & Stanghellini, G. (2016). How to Link Brain and Experience? Spatiotemporal Psychopathology of the Lived Body. *Frontiers in Human Neuroscience*, *10*(April), 1–15.
<https://doi.org/10.3389/fnhum.2016.00172>
- Northoff, G., Wainio-Theberge, S., & Evers, K. (2020a). Is temporo-spatial dynamics the “common currency” of brain and mind? In Quest of “Spatiotemporal Neuroscience.” *Physics of Life Reviews*, *33*, 34–54. <https://doi.org/10.1016/j.plrev.2019.05.002>
- Northoff, G., Wainio-Theberge, S., & Evers, K. (2020b). Spatiotemporal neuroscience – what is it and why we need it. *Physics of Life Reviews*, *33*, 78–87. <https://doi.org/10.1016/j.plrev.2020.06.005>
- Omidvarnia, A., Zalesky, A., Mansour L, S., Van De Ville, D., Jackson, G. D., & Pedersen, M. (2021). Temporal complexity of fMRI is reproducible and correlates with higher order cognition. *NeuroImage*, *230*(October 2020), 117760. <https://doi.org/10.1016/j.neuroimage.2021.117760>
- Parvizi, J., & Damasio, A. (2001). Consciousness and the brainstem. *Cognition*, *79*(1–2), 135–160.
[https://doi.org/10.1016/S0010-0277\(00\)00127-X](https://doi.org/10.1016/S0010-0277(00)00127-X)
- Pedersen, M., Omidvarnia, A., Walz, J. M., Zalesky, A., & Jackson, G. D. (2017). Spontaneous brain network activity: Analysis of its temporal complexity. *Network Neuroscience*, *1*(2), 100–115.

https://doi.org/10.1162/netn_a_00006

- Pedersen, M., Omidvarnia, A., Zalesky, A., & Jackson, G. D. (2018a). On the relationship between instantaneous phase synchrony and correlation-based sliding windows for time-resolved fMRI connectivity analysis. *NeuroImage*, *181*(June), 85–94.
<https://doi.org/10.1016/j.neuroimage.2018.06.020>
- Pedersen, M., Omidvarnia, A., Zalesky, A., & Jackson, G. D. (2018b). On the relationship between instantaneous phase synchrony and correlation-based sliding windows for time-resolved fMRI connectivity analysis. *NeuroImage*, *181*(April), 85–94.
<https://doi.org/10.1016/j.neuroimage.2018.06.020>
- Preti, M. G., Bolton, T. A., & Van De Ville, D. (2017). The dynamic functional connectome: State-of-the-art and perspectives. *NeuroImage*, *160*(December 2016), 41–54.
<https://doi.org/10.1016/j.neuroimage.2016.12.061>
- Rabinovich, M. I., Huerta, R., Varona, P., & Afraimovich, V. S. (2008). Transient cognitive dynamics, metastability, and decision making. *PLoS Computational Biology*, *4*(5), 25–30.
<https://doi.org/10.1371/journal.pcbi.1000072>
- Richman, J. S., & Moorman, J. R. (2000). Physiological time-series analysis using approximate entropy and sample entropy Physiological time-series analysis using approximate entropy and sample entropy. *Cardiovascular Research*, 2039–2049.
- Shine, J. M., Breakspear, M., Bell, P. T., Ehgoetz Martens, K., Shine, R., Koyejo, O., ... Poldrack, R. A. (2019). Human cognition involves the dynamic integration of neural activity and neuromodulatory systems. *Nature Neuroscience*, *22*(2), 289–296.
<https://doi.org/10.1038/s41593-018-0312-0>
- Smith, S. M., Fox, P. T., Miller, K. L., Glahn, D. C., Fox, P. M., Mackay, C. E., ... Beckmann, C. F. (2009). Correspondence of the brain's functional architecture during activation and rest. *Proceedings of the National Academy of Sciences of the United States of America*, *106*(31), 13040–13045.
<https://doi.org/10.1073/pnas.0905267106>
- Tagliazucchi, E., Von Wegner, F., Morzelewski, A., Brodbeck, V., Jahnke, K., & Laufs, H. (2013). Breakdown of long-range temporal dependence in default mode and attention networks during deep sleep. *Proceedings of the National Academy of Sciences of the United States of America*, *110*(38), 15419–15424. <https://doi.org/10.1073/pnas.1312848110>
- Tanabe, S., Huang, Z., Zhang, J., Chen, Y., Fogel, S., Doyon, J., ... Northoff, G. (2020). Altered Global Brain Signal during Physiologic, Pharmacologic, and Pathologic States of Unconsciousness in Humans and Rats. *Anesthesiology*, *6*(6), 1392–1406.
<https://doi.org/10.1097/ALN.0000000000003197>
- Tognoli, E., & Kelso, J. A. S. (2014). The Metastable Brain. *Neuron*, *81*(1), 35–48.
<https://doi.org/10.1016/j.neuron.2013.12.022>
- Tononi, G., Boly, M., Massimini, M., & Koch, C. (2016). Integrated information theory: From consciousness to its physical substrate. *Nature Reviews Neuroscience*, *17*(7), 450–461.
<https://doi.org/10.1038/nrn.2016.44>
- Varley, T. F., Denny, V., Sporns, O., & Patania, A. (2021). Topological analysis of differential effects of ketamine and propofol anaesthesia on brain dynamics. *Royal Society Open Science*, *8*(6).
<https://doi.org/10.1098/rsos.201971>
- Varley, T. F., Luppi, A. I., Pappas, I., Naci, L., Adapa, R., Owen, A. M., ... Stamatakis, E. A. (2020). Consciousness & Brain Functional Complexity in Propofol Anaesthesia. *Scientific Reports*, *10*(1),

1–13. <https://doi.org/10.1038/s41598-020-57695-3>

Wang, Z., Li, Y., Childress, A. R., & Detre, J. A. (2014). Brain entropy mapping using fMRI. *PLoS ONE*, 9(3), 1–8. <https://doi.org/10.1371/journal.pone.0089948>

Yoshimi, J., & Vinson, D. W. (2015). Extending Gurwitsch's field theory of consciousness. *Consciousness and Cognition*, 34, 104–123. <https://doi.org/10.1016/j.concog.2015.03.017>

Zhang, J., Huang, Z., Chen, Y., Zhang, J., Ghinda, D., Nikolova, Y., ... Northoff, G. (2018). Breakdown in the temporal and spatial organization of spontaneous brain activity during general anesthesia. *Human Brain Mapping*, 39(5), 2035–2046. <https://doi.org/10.1002/hbm.23984>

Zhang, S., Rogers, B. P., Morgan, V. L., & Chang, C. (2020). Association between fMRI brain entropy features and behavioral measures. In *PROCEEDINGS OF SPIE* (p. 30). <https://doi.org/10.1117/12.2549342>

Zilio, F., Gomez-Pilar, J., Cao, S., Zhang, J., Zang, D., Qi, Z., ... Northoff, G. (2021). Are intrinsic neural timescales related to sensory processing? Evidence from abnormal behavioral states. *NeuroImage*, 226(November 2020), 117579. <https://doi.org/10.1016/j.neuroimage.2020.117579>

REVIEWERS' COMMENTS:

Reviewer #1 (Remarks to the Author):

all my comments were well addressed so the paper can be accepted for publication

Reviewer #2 (Remarks to the Author):

The authors have adequately addressed all my concerns and I have no further comments.

Reviewer #3 (Remarks to the Author):

The authors have adequately addressed my comments in the revised manuscript.